# AP-OOD: Attention Pooling for Out-of-Distribution Detection

**Claus Hofmann**[1]     **Christian Huber**[2]     **Bernhard Lehner**[2]
**Daniel Klotz**[3]     **Sepp Hochreiter**[1]     **Werner Zellinger**[4]

[1] Institute for Machine Learning, JKU LIT SAL IWS Lab,
Johannes Kepler University, Linz, Austria
[2] Silicon Austria Labs, JKU LIT SAL IWS Lab, Linz, Austria
[3] Interdisciplinary Transformation University Austria, Linz, Austria
[4] ELLIS Unit, LIT AI Lab, Institute for Machine Learning, JKU Linz, Austria

## Abstract

Out-of-distribution (OOD) detection, which maps high-dimensional data into a scalar OOD score, is critical for the reliable deployment of machine learning models. A key challenge in recent research is how to effectively leverage and aggregate token embeddings from language models to obtain the OOD score. In this work, we propose AP-OOD, a novel OOD detection method for natural language that goes beyond simple average-based aggregation by exploiting token-level information. AP-OOD is a semi-supervised approach that flexibly interpolates between unsupervised and supervised settings, enabling the use of limited auxiliary outlier data. Empirically, AP-OOD sets a new state of the art in OOD detection for text: in the unsupervised setting, it reduces the FPR95 (false positive rate at 95% true positives) from 27.84% to 4.67% on XSUM summarization, and from 77.08% to 70.37% on WMT15 En–Fr translation. Code is available at https://github.com/ml-jku/ap-ood.

## 1 Introduction

Out-of-distribution (OOD) detection is essential for deploying machine learning models in the real world. In practical settings many models encounter inputs that deviate from the model's training distribution. For example, a language model trained to summarize news articles might also receive a prompt with a cooking recipe. In such situations, models may assign unwarranted confidence to their predictions, leading to erroneous outputs and hallucination. A hallucination is a state in which the model generates output that is nonsensical or unfaithful to the prompt (Farquhar et al., 2024). For example, Ren et al. (2023) observe that a common failure case in abstractive summarization is for the model to output "All images are copyrighted" when prompted to summarize news articles from a publisher (CNN) that differs from what it was trained on (BBC). Many authors attribute hallucination to model uncertainty (e.g., Yadkori et al., 2024; Aichberger et al., 2025), which decomposes into aleatoric uncertainty (resulting from noise in the data) and epistemic uncertainty (resulting from a lack of training data). OOD prompts exhibit high epistemic uncertainty (Ren et al., 2023). The purpose of OOD detection is to classify these inputs as OOD such that the system can then, for instance, notify the user that no output can be generated. Many existing post-hoc OOD detection methods (e.g., Huang et al., 2021; Sun & Li, 2022; Wang et al., 2022) assume a classifier as the base model. In contrast, in language modeling, the base model is typically an autoregressive generative model without an explicit classification head. This necessitates the development of OOD detection methods specifically tailored for language modeling. Our contributions are as follows:

1. We propose AP-OOD, an OOD detection approach for natural language that leverages token-level information to detect OOD sequences.

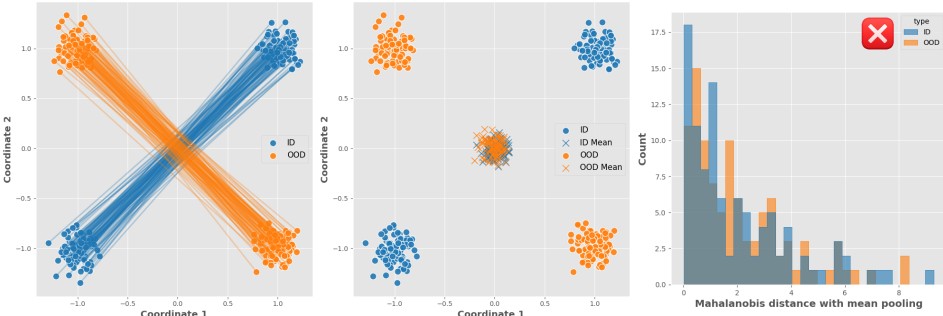

Figure 1: Illustrative example for the failure of mean pooling. (**Left**) ID and OOD sequences $\boldsymbol{Z}_i \in \mathbb{R}^{2\times 2}$, where each sequence contains a pair of token embeddings with two features each. Token embeddings that belong to the same sequence are connected with lines. (**Center**) The means of the ID and OOD sequences both cluster around the origin. (**Right**) A mean pooling approach cannot discriminate between the ID and OOD sequences.

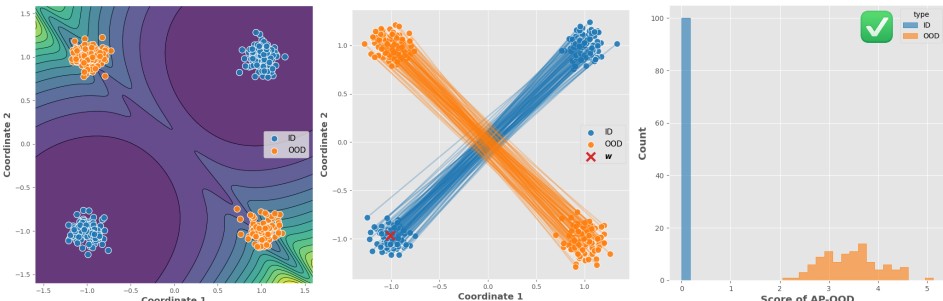

Figure 2: Illustrative example for the mechanism that AP-OOD uses to correctly discriminate between ID and OOD (as opposed to the mean pooling approaches). The setting is the same as in Figure 1. (**Left**) The loss landscape forms two basins at the locations of the ID token embeddings. (**Center**) After training AP-OOD with a single weight vector $\boldsymbol{w}$, the learned $\boldsymbol{w}$ is located in one of the basins. (**Right**) AP-OOD achieves perfect discrimination between the ID and OOD sequences.

2. AP-OOD is a semi-supervised approach: It can be applied in unsupervised (i.e., when there exists no knowledge about OOD samples) and supervised settings (i.e., when some OOD data of interest is available to the practitioner), and smoothly interpolates between the two.

3. We show that AP-OOD can improve OOD detection for natural language in summarization and translation.

4. We provide a theoretical motivation for the suitability of AP-OOD for OOD detection on tokenized data.

## 1.1 BACKGROUND

Conditional language models are typically trained given input sequences $(\boldsymbol{x}_1, \boldsymbol{x}_2, \ldots, \boldsymbol{x}_N)$ with $\boldsymbol{x}_i \in \mathcal{X}$[1] to autoregressively generate target sequences $(\boldsymbol{y}_1, \boldsymbol{y}_2, \ldots, \boldsymbol{y}_N)$ with $\boldsymbol{y}_i \in \mathcal{X}$. The input sequences are drawn i.i.d.: $\boldsymbol{x}_i \sim p_{\text{ID}}$. We consider input sequences $\boldsymbol{x} \in \mathcal{X}$ that deviate considerably from the data generation $p_{\text{ID}}(\boldsymbol{x})$ that defines the "normality" of our data as OOD. Following Ruff et al. (2021), an observed sequence is OOD if it is an element of the set

$$\mathbb{O} := \{\boldsymbol{x} \in \mathcal{X} \mid p_{\text{ID}}(\boldsymbol{x}) < \epsilon\} \text{ where } \epsilon \geq 0, \tag{1}$$

and $\epsilon \in \mathbb{R}$ is a density threshold. In practice, it is common (e.g., Hendrycks & Gimpel, 2016; Lee et al., 2018; Hofmann et al., 2024) to define a score $s : \mathcal{Z} \to \mathbb{R}$ that uses an encoder

---

[1]We use $\mathcal{X} := \bigcup_{S \geq 1} \mathcal{V}^S$ for the set of input sequences, and $\mathcal{V} := \{v_1, \ldots, v_V\}$ is the vocabulary.

$\phi : \mathcal{X} \to \mathcal{Z}$ (where $\mathcal{Z}$ denotes an embedding space). Given $s$ and $\phi$, OOD detection can be formulated as a binary classification task with the classes in-distribution (ID) and OOD:

$$\hat{B}(\boldsymbol{x}, \gamma) \;=\; \begin{cases} \text{ID} & \text{if } s(\phi(\boldsymbol{x})) \geq \gamma \\ \text{OOD} & \text{if } s(\phi(\boldsymbol{x})) < \gamma \end{cases}. \tag{2}$$

The outlier score should — in the best case — preserve the density ranking, but it does not have to fulfill all requirements of a probability density (proper normalization or nonnegativity). For evaluation, the threshold $\gamma \in \mathbb{R}$ is typically chosen such that 95% of ID samples from a previously unseen validation set are correctly classified as ID. However, metrics like the area under the receiver operating characteristic (AUROC) can be directly computed on $s(\phi(\boldsymbol{x}))$ without fixing $\gamma$, since the AUROC sweeps over all possible thresholds.

## 2 Method

AP-OOD is a semi-supervised method: It can be trained without access to outlier data (unsupervised), and with access to outlier data (supervised), and can smoothly transition between those two scenarios as more outlier data becomes available for training. In the following, we first introduce AP-OOD in an unsupervised scenario (Section 2.1) and generalize it to the supervised scenario (Section 2.2).

### 2.1 Unsupervised OOD Detection

**Background** Ren et al. (2023) propose to detect OOD inputs using token embeddings obtained from a transformer encoder–decoder model (Vaswani et al., 2017b) trained on the language modeling task. Given an input sequence $\boldsymbol{x}$, they obtain a sequence of token embeddings. They compare obtaining embeddings $\boldsymbol{E} \in \mathcal{Z}^2$ from the encoder $\phi_{\text{enc}} : \mathcal{X} \to \mathcal{Z}$ and generating a sequence of embeddings $\boldsymbol{G} \in \mathcal{Z}$ using the decoder $\phi_{\text{dec}} : \mathcal{Z} \to \mathcal{Z}$:

$$\boldsymbol{E} \;:=\; \phi_{\text{enc}}(\boldsymbol{x}) \qquad \boldsymbol{G} \;:=\; \phi_{\text{dec}}(\boldsymbol{E}). \tag{3}$$

For clarity, we write $\boldsymbol{Z} \in \mathcal{Z}$ for a sequence of token embeddings, whether produced by the encoder or the decoder, and we call $\boldsymbol{Z}$ the sequence representation of $\boldsymbol{x}$. To obtain a single vector $\bar{\boldsymbol{z}} \in \mathbb{R}^D$, Ren et al. (2023) perform mean pooling:

$$\bar{\boldsymbol{z}} \;:=\; \frac{1}{S} \sum_{s=1}^{S} \boldsymbol{z}_s. \tag{4}$$

Then, they propose to measure whether $\bar{\boldsymbol{z}}$ is OOD by first fitting a Gaussian distribution $\mathcal{N}(\boldsymbol{\mu}, \boldsymbol{\Sigma})$, $\boldsymbol{\mu} \in \mathbb{R}^D$, $\boldsymbol{\Sigma} \in \mathbb{R}^{D \times D}$ to the per-sequence mean embeddings computed from the training corpus, and then computing the squared Mahalanobis distance between $\bar{\boldsymbol{z}}$ and $\boldsymbol{\mu}$:

$$d_{\text{Maha}}^2(\bar{\boldsymbol{z}}, \boldsymbol{\mu}) \;:=\; (\bar{\boldsymbol{z}} - \boldsymbol{\mu})^T \boldsymbol{\Sigma}^{-1}(\bar{\boldsymbol{z}} - \boldsymbol{\mu}) \quad \text{and} \quad s_{\text{Maha}}(\bar{\boldsymbol{z}}) \;:=\; - d_{\text{Maha}}^2(\bar{\boldsymbol{z}}, \boldsymbol{\mu}). \tag{5}$$

**Averaging hides anomalies.** The key limitation of the approach described above is the use of the **mean** of the token embeddings $\boldsymbol{Z}$: Averaging the entire sequence into the mean $\bar{\boldsymbol{z}}$ discards the token-level structure that would otherwise be informative for detecting whether a sequence is OOD. Figure 1 shows a toy example of this failure mode: The ID and OOD sequences are indistinguishable using their means, and therefore, the Mahalanobis distance with mean pooling fails to discriminate between them.

**Mahalanobis decomposition.** To address this limitation, we begin by expressing the Mahalanobis distance as a directional decomposition:

$$d_{\text{Maha}}^2(\bar{\boldsymbol{z}}, \boldsymbol{\mu}) \;=\; \sum_{j=1}^{D} \left( \boldsymbol{w}_j^T \bar{\boldsymbol{z}} - \boldsymbol{w}_j^T \boldsymbol{\mu} \right)^2, \tag{6}$$

The weight vectors $\boldsymbol{w}_j \in \mathbb{R}^D$ form a basis of $\mathbb{R}^D$ and determine $\boldsymbol{\Sigma}^{-1}$ via $\boldsymbol{\Sigma}^{-1} = \sum_{j=1}^{D} \boldsymbol{w}_j \boldsymbol{w}_j^T$. One possibility to map a given $\boldsymbol{\Sigma}^{-1}$ to weight vectors $\boldsymbol{w}_j$ is to select the directions of the $\boldsymbol{w}_j$ as the unit-norm eigenvectors of $\boldsymbol{\Sigma}^{-1}$, and to select the squared norms of the $\boldsymbol{w}_j$ as their corresponding eigenvalues (see Appendix B.2).

---

[2] We use $\mathcal{Z} := \bigcup_{S \geq 1} \mathbb{R}^{D \times S}$ for all finite-length sequences of $D$-dimensional token embeddings.

---

**Algorithm 1** AP-OOD

---

**Require:** $(\boldsymbol{x}_1, \ldots, \boldsymbol{x}_N)$, $\phi_{\text{enc}}$, $\phi_{\text{dec}}$, $\beta$, $M$, nsteps

1: **for** $i = 1$ to $N$ **do**
2:      Compute sequence embedding $\boldsymbol{Z}_i$ using $\boldsymbol{Z}_i \leftarrow \phi_{\text{enc}}(\boldsymbol{x}_i)$ or $\boldsymbol{Z}_i \leftarrow \phi_{\text{dec}}(\phi_{\text{enc}}(\boldsymbol{x}_i))$.
3: **for** step $= 1$ to nsteps **do**
4:      Randomly sample mini-batch indices $\mathcal{B} \subset \{1, \ldots, N\}$
5:      Collect mini-batch $\{\boldsymbol{Z}_i\}_{i \in \mathcal{B}}$.
6:      Form batch-local concatenation $\tilde{\boldsymbol{Z}}_B \leftarrow \|_{i \in \mathcal{B}} \boldsymbol{Z}_i$.
7:      Compute loss $\mathcal{L} \leftarrow \frac{1}{|\mathcal{B}|} \sum_{i \in \mathcal{B}} d^2(\boldsymbol{Z}_i, \tilde{\boldsymbol{Z}}_B) - \sum_{j=1}^{M} \log(\|\boldsymbol{w}_j\|_2^2)$.
8:      Compute gradients of $\mathcal{L}$ w.r.t. $(\boldsymbol{w}_1, \ldots, \boldsymbol{w}_M)$ and perform a gradient update
9: Do mini-batch attention pooling to compute $\boldsymbol{\mu}_j \leftarrow \tilde{\boldsymbol{Z}} \text{softmax}(\beta \, \tilde{\boldsymbol{Z}}^T \boldsymbol{w}_j)$ (Appendix C.1)
10: $s(\boldsymbol{Z}) \leftarrow \sum_{j=1}^{M} -d_j^2(\boldsymbol{Z}, \tilde{\boldsymbol{Z}}) + \log\left(\|\boldsymbol{w}_j\|_2^2\right)$.
11: **return** $s(\cdot)$

---

**Beyond mean pooling.** To overcome the limitations of mean pooling, we generalize Equation (6) by using attention pooling (Bahdanau, 2014; Ramsauer et al., 2021):

$$\text{AttPool}_\beta(\boldsymbol{Z}, \boldsymbol{w}) := \boldsymbol{Z}\text{softmax}(\beta \, \boldsymbol{Z}^T \boldsymbol{w}) \quad \text{and} \quad \bar{\boldsymbol{z}} := \text{AttPool}_\beta(\boldsymbol{Z}, \boldsymbol{w}). \tag{7}$$

where $\beta \in \mathbb{R}_{\geq 0}$ is the inverse temperature, and $\boldsymbol{w} \in \mathbb{R}^D$ is a learnable query. AP-OOD also uses attention for the corpus-wide pooling: Starting with the sequence representations $(\boldsymbol{Z}_1, \ldots, \boldsymbol{Z}_N)$ with $\boldsymbol{Z}_i := \phi_{\text{enc}}(\boldsymbol{x}_i)$, we construct $\tilde{\boldsymbol{Z}} \in \mathcal{Z}$ by concatenating the sequence representations: $\tilde{\boldsymbol{Z}} := (\boldsymbol{Z}_1 \| \cdots \| \boldsymbol{Z}_N)$. AP-OOD computes the global prototype using $\boldsymbol{\mu} := \text{AttPool}_\beta(\tilde{\boldsymbol{Z}}, \boldsymbol{w})$. Given the $\bar{\boldsymbol{z}}$ and $\boldsymbol{\mu}$ from the attention pooling, AP-OOD computes the squared distance $d^2(\boldsymbol{Z}, \tilde{\boldsymbol{Z}})$ analogous to Equation (6):

$$d^2(\boldsymbol{Z}, \tilde{\boldsymbol{Z}}) := \sum_{j=1}^{M} \left(\boldsymbol{w}_j^T \boldsymbol{Z}\text{softmax}(\beta \, \boldsymbol{Z}^T \boldsymbol{w}_j) - \boldsymbol{w}_j^T \tilde{\boldsymbol{Z}}\text{softmax}(\beta \, \tilde{\boldsymbol{Z}}^T \boldsymbol{w}_j)\right)^2 = \sum_{j=1}^{M} d_j^2(\boldsymbol{Z}, \tilde{\boldsymbol{Z}}). \tag{8}$$

We refer to $M \in \mathbb{N}$ as the number of heads. In general, $M$ does not need to equal the embedding dimension $D$. We show in Appendix B.4 that, when $\beta = 0$ and $M = D$, Equation (8) reduces to the Mahalanobis distance (Equations (5) and (6)). To the best of our knowledge, AP-OOD is the first approach to integrate attention pooling into the Mahalanobis distance via a learnable directional decomposition. In Appendix B.1, we show that $s_{\min}(\boldsymbol{Z}) = \min_j -d_j^2(\boldsymbol{Z}, \tilde{\boldsymbol{Z}}) + \log(\|\boldsymbol{w}_j\|_2^2)$ is a score function as defined in Equation (2). Our score arises naturally as the upper bound

$$s(\boldsymbol{Z}) := \sum_{j=1}^{M} -d_j^2(\boldsymbol{Z}, \tilde{\boldsymbol{Z}}) + \log(\|\boldsymbol{w}_j\|_2^2) = -d^2(\boldsymbol{Z}, \tilde{\boldsymbol{Z}}) + \sum_{j=1}^{M} \log(\|\boldsymbol{w}_j\|_2^2). \tag{9}$$

In Appendix D.7, we empirically compare the min-based score $s_{\min}(\boldsymbol{Z})$ to its upper-bound variant $s(\boldsymbol{Z})$ and find that $s(\boldsymbol{Z})$ yields stronger OOD discrimination. The choice of this score naturally leads to the loss function of AP-OOD:

$$\mathcal{L}(\boldsymbol{w}_1, \ldots, \boldsymbol{w}_M) := \frac{1}{N} \sum_{i=1}^{N} d^2(\boldsymbol{Z}_i, \tilde{\boldsymbol{Z}}) - \sum_{j=1}^{M} \log\left(\|\boldsymbol{w}_j\|_2^2\right). \tag{10}$$

We provide the pseudocode for AP-OOD in Algorithm 1. Scaling to large data sets requires efficient computation of $\boldsymbol{\mu} = \tilde{\boldsymbol{Z}}\text{softmax}(\beta \, \tilde{\boldsymbol{Z}}^T \boldsymbol{w})$; the naive method loads the entire concatenated sequence $\tilde{\boldsymbol{Z}}$ into memory, but we reduce the memory footprint by performing attention pooling on mini-batches. We describe this procedure in Appendix C.1.

**Multiple queries per head.** We now extend AP-OOD and use multiple queries per head. We use a set of stacked queries $\boldsymbol{W}_j = (\boldsymbol{w}_{j1}, \ldots, \boldsymbol{w}_{jT}) \in \mathbb{R}^{D \times T}$ per head. For simplicity, we consider a single head with the queries $\boldsymbol{W} \in \mathbb{R}^{D \times T}$ for now. We begin by extending the

softmax notation from Ramsauer et al. (2021) to matrix-valued arguments. Given a matrix $\boldsymbol{A} \in \mathbb{R}^{S \times T}$

$$\text{softmax}(\beta \boldsymbol{A})_{st} := \frac{\exp(\beta a_{st})}{\sum_{s'=1}^{S} \sum_{t'=1}^{T} \exp(\beta a_{s't'})}. \tag{11}$$

In other words, the softmax normalizes over the rows and columns of $\boldsymbol{A}$. Next, we extend the attention pooling process from Equation (7) with the matrix-valued softmax: AP-OOD transforms the sequence representation $\boldsymbol{Z} \in \mathbb{R}^{D \times S}$ with $S$ tokens to a new sequence representation $\bar{\boldsymbol{Z}} \in \mathbb{R}^{D \times T}$ with $T$ tokens using $\bar{\boldsymbol{Z}} := \boldsymbol{Z}\boldsymbol{P}$. The updated attention pooling process is

$$\text{AttPool}_\beta(\boldsymbol{Z}, \boldsymbol{W}) := \boldsymbol{Z}\text{softmax}(\beta \boldsymbol{Z}^T \boldsymbol{W}) \quad \text{and} \quad \bar{\boldsymbol{Z}} := \text{AttPool}_\beta(\boldsymbol{Z}, \boldsymbol{W}). \tag{12}$$

To the best of our knowledge, this work is the first to use a matrix-valued global softmax to transform a sequence $\boldsymbol{Z}$ into another sequence $\bar{\boldsymbol{Z}}$, though similar matrix-based memory updates have recently been introduced in recurrent architectures (Beck et al., 2024). Finally, AP-OOD uses $\boldsymbol{W} \in \mathbb{R}^{D \times T}$ to transform the $\bar{\boldsymbol{Z}} \in \mathbb{R}^{D \times T}$ to a scalar using $\langle \boldsymbol{W}, \bar{\boldsymbol{Z}} \rangle_F = \text{vec}(\boldsymbol{W})^T \text{vec}(\bar{\boldsymbol{Z}}) = \text{Tr}(\boldsymbol{W}^T \bar{\boldsymbol{Z}})$ (where $\langle \cdot, \cdot \rangle_F$ denotes the Frobenius inner product, and $\text{vec}(\cdot)$ denotes the flattening operation). To summarize, the extended squared distance is

$$d^2(\boldsymbol{Z}, \tilde{\boldsymbol{Z}}) := \sum_{j=1}^{M} \left( \text{Tr}(\boldsymbol{W}_j^T \boldsymbol{Z}\text{softmax}(\beta \boldsymbol{Z}^T \boldsymbol{W}_j)) - \text{Tr}(\boldsymbol{W}_j^T \tilde{\boldsymbol{Z}}\text{softmax}(\beta \tilde{\boldsymbol{Z}}^T \boldsymbol{W}_j)) \right)^2. \tag{13}$$

Finally, the regularizing term is $-\log(||\boldsymbol{W}||_F^2)$ (where $|| \cdot ||_F^2$ denotes the squared Frobenius norm). To summarize, the extended loss is

$$\mathcal{L}(\boldsymbol{W}_1, \ldots, \boldsymbol{W}_M) := \frac{1}{N} \sum_{i=1}^{N} d^2(\boldsymbol{Z}_i, \tilde{\boldsymbol{Z}}) - \sum_{j=1}^{M} \log\left(||\boldsymbol{W}_j||_F^2\right). \tag{14}$$

We provide PyTorch-style pseudocode implementing Equation (14) in Appendix C.2, and we analyze Equation (13) through the lens of kernel functions in Appendix B.3.

## 2.2 Supervised OOD Detection

**Background.** Supplying an OOD detector with information about the distribution of the OOD examples at training time can improve the ID–OOD decision boundary (Hendrycks et al., 2018). In practice, it is hard to find OOD data for training that is fully indicative of the OOD distribution seen during inference. Outlier exposure (OE; Hendrycks et al., 2018) therefore uses a large and diverse auxiliary outlier set (AUX; e.g., C4 for text data) as a stand-in for the OOD case. However, acquiring such large and diverse AUX datasets is not always possible. For example, consider a translation task with a less widely spoken source language. As another example, consider detecting defects in industrial machines using recordings of their sounds (Nishida et al., 2024). Practitioners can collect a relatively large amount of ID audio data from machines while they run without defects. However, it is much harder to collect diverse AUX examples from defective machines because defects are infrequent. In such a case, one might have to resort to a smaller AUX data set. Therefore, an OOD detector should scale gracefully with the degree of auxiliary supervision, adapting to the available number of AUX examples (e.g., Ruff et al., 2019; Liznerski et al., 2022; Yoon et al., 2023; Ivanov et al., 2024; Qiao et al., 2024).

**Utilizing AUX data.** To adapt AP-OOD to the supervised setting, we follow Ruff et al. (2019) and Liznerski et al. (2022): AP-OOD punishes large squared distances $d^2(\boldsymbol{Z}, \tilde{\boldsymbol{Z}})$ for ID samples $\boldsymbol{Z}$ and encourages large squared distances for AUX samples $\boldsymbol{Z}$. Formally, AP-OOD minimizes the binary cross-entropy loss with the classes ID and AUX with $p(y = \text{ID}|\boldsymbol{Z}) = \exp(-d^2(\boldsymbol{Z}, \tilde{\boldsymbol{Z}}))$. Given $N$ ID examples $(\boldsymbol{Z}_1, \ldots, \boldsymbol{Z}_N)$, and $N'$ AUX examples $(\boldsymbol{Z}_{N+1}, \ldots, \boldsymbol{Z}_{N+N'})$, AP-OOD minimizes the supervised loss

$$\mathcal{L}_{\text{SUP}} := \frac{1}{N + N'} \sum_{i=1}^{N} d^2(\boldsymbol{Z}_i, \tilde{\boldsymbol{Z}}) - \lambda \frac{1}{N + N'} \sum_{i=N+1}^{N+N'} \log(1 - \exp(-d^2(\boldsymbol{Z}_i, \tilde{\boldsymbol{Z}}))), \tag{15}$$

where $\lambda \in \mathbb{R}_{\geq 0}$. If $\lambda = 0$, $\mathcal{L}_{\text{SUP}}$ equals the unsupervised loss $\mathcal{L}$ without the regularizing term.

Table 1: Unsupervised OOD detection performance on text summarization. We compare results from AP-OOD, Mahalanobis (Lee et al., 2018; Ren et al., 2023), KNN (Sun et al., 2022), Deep SVDD (Ruff et al., 2018), model perplexity (Ren et al., 2023), and entropy (Malinin & Gales, 2020) on PEGASUS$_{\text{LARGE}}$ trained on XSUM as the ID data set. ↓ indicates "lower is better" and ↑ "higher is better". All values in %. We estimate standard deviations across five independent data set splits and training runs.

| | | CNN/DM | Newsroom | Reddit | Samsum | Mean |
|---|---|---|---|---|---|---|
| **Input OOD** | | | | | | |
| Mahalanobis | AUROC ↑ | $68.95^{\pm 0.20}$ | $86.40^{\pm 0.14}$ | $98.64^{\pm 0.02}$ | $\mathbf{99.77^{\pm 0.02}}$ | 88.44 |
| | FPR95 ↓ | $92.16^{\pm 0.20}$ | $64.02^{\pm 0.51}$ | $2.43^{\pm 0.12}$ | $\underline{0.17^{\pm 0.02}}$ | 39.69 |
| KNN | AUROC ↑ | $54.32^{\pm 0.16}$ | $73.83^{\pm 0.14}$ | $94.53^{\pm 0.10}$ | $98.84^{\pm 0.01}$ | 80.38 |
| | FPR95 ↓ | $99.39^{\pm 0.03}$ | $88.49^{\pm 0.34}$ | $51.26^{\pm 0.89}$ | $3.09^{\pm 0.12}$ | 60.56 |
| Deep SVDD | AUROC ↑ | $\underline{75.75^{\pm 0.86}}$ | $\underline{91.36^{\pm 0.38}}$ | $\underline{99.71^{\pm 0.08}}$ | $99.57^{\pm 0.08}$ | $\underline{91.60}$ |
| | FPR95 ↓ | $\underline{74.20^{\pm 1.60}}$ | $\underline{36.05^{\pm 1.71}}$ | $\underline{0.39^{\pm 0.23}}$ | $0.72^{\pm 0.28}$ | $\underline{27.84}$ |
| AP-OOD (Ours) | AUROC ↑ | $\mathbf{97.48^{\pm 0.32}}$ | $\mathbf{98.54^{\pm 0.10}}$ | $\mathbf{99.88^{\pm 0.05}}$ | $99.76^{\pm 0.05}$ | $\mathbf{98.91}$ |
| | FPR95 ↓ | $\mathbf{12.88^{\pm 1.68}}$ | $\mathbf{5.78^{\pm 0.58}}$ | $\mathbf{0.00^{\pm 0.00}}$ | $\mathbf{0.00^{\pm 0.01}}$ | $\mathbf{4.67}$ |
| **Output OOD** | | | | | | |
| Perplexity | AUROC ↑ | $41.65^{\pm 0.17}$ | $52.86^{\pm 0.39}$ | $82.59^{\pm 0.37}$ | $77.90^{\pm 0.08}$ | 63.75 |
| | FPR95 ↓ | $77.83^{\pm 0.16}$ | $79.40^{\pm 0.41}$ | $48.25^{\pm 1.05}$ | $47.35^{\pm 0.26}$ | 63.21 |
| Entropy | AUROC ↑ | $59.76^{\pm 0.10}$ | $76.92^{\pm 0.08}$ | $93.42^{\pm 0.19}$ | $87.07^{\pm 0.15}$ | 79.30 |
| | FPR95 ↓ | $79.26^{\pm 0.66}$ | $64.65^{\pm 0.64}$ | $30.45^{\pm 1.00}$ | $50.83^{\pm 0.65}$ | 56.30 |
| Mahalanobis | AUROC ↑ | $62.67^{\pm 0.35}$ | $87.38^{\pm 0.06}$ | $97.13^{\pm 0.10}$ | $96.99^{\pm 0.03}$ | 86.04 |
| | FPR95 ↓ | $89.46^{\pm 0.19}$ | $49.06^{\pm 0.51}$ | $12.27^{\pm 0.53}$ | $15.25^{\pm 0.33}$ | 41.51 |
| KNN | AUROC ↑ | $\underline{74.17^{\pm 0.13}}$ | $86.69^{\pm 0.12}$ | $95.82^{\pm 0.12}$ | $\underline{97.28^{\pm 0.05}}$ | $\underline{88.49}$ |
| | FPR95 ↓ | $\underline{73.36^{\pm 0.15}}$ | $54.30^{\pm 0.36}$ | $17.28^{\pm 0.73}$ | $\underline{10.99^{\pm 0.36}}$ | 38.98 |
| Deep SVDD | AUROC ↑ | $66.59^{\pm 1.25}$ | $\underline{93.47^{\pm 0.31}}$ | $\underline{97.58^{\pm 0.24}}$ | $95.61^{\pm 0.21}$ | 88.31 |
| | FPR95 ↓ | $77.67^{\pm 1.34}$ | $\mathbf{20.67^{\pm 0.62}}$ | $\underline{9.31^{\pm 1.24}}$ | $21.66^{\pm 1.39}$ | $\underline{32.33}$ |
| AP-OOD (Ours) | AUROC ↑ | $\mathbf{93.53^{\pm 0.34}}$ | $\mathbf{94.41^{\pm 0.28}}$ | $\mathbf{98.72^{\pm 0.41}}$ | $\mathbf{98.89^{\pm 0.09}}$ | $\mathbf{96.39}$ |
| | FPR95 ↓ | $\mathbf{29.58^{\pm 2.59}}$ | $\underline{28.18^{\pm 1.78}}$ | $\mathbf{3.18^{\pm 2.36}}$ | $\mathbf{4.09^{\pm 0.88}}$ | $\mathbf{16.26}$ |

## 3 EXPERIMENTS

**Toy experiment.** We present a toy experiment illustrating the main intuitions behind AP-OOD. Figure 1 demonstrates a simple failure mode of mean pooling approaches: First, we generate ID and OOD token embeddings $\boldsymbol{Z}_i \in \mathbb{R}^{2 \times 2}$. Each ID sequence representation consists of one token sampled from $\mathcal{N}((1,1), \sigma^2\boldsymbol{I})$ (where $\sigma := 0.1$) and one token sampled from $\mathcal{N}((-1,-1), \sigma^2\boldsymbol{I})$. The OOD sequences contain two tokens sampled from $\mathcal{N}((-1,1), \sigma^2\boldsymbol{I})$ and $\mathcal{N}((1,-1), \sigma^2\boldsymbol{I})$, respectively. The left panel shows the generated sequences, where each sequence consists of two dots (representing the two tokens) connected by a line. Because the means of the ID and OOD sequences both cluster around the origin (central panel), the Mahalanobis distance with mean pooling fails to discriminate between them (right panel). Figure 2 shows how AP-OOD overcomes this limitation: We set $M = 1$ and $T = 1$ and train AP-OOD as described in Section 2.1 on the ID data only, but we modify the pooling mechanism from Equation (7): We replace the dot product similarity in the softmax with the negative squared Euclidean distance, as it is known to work better in low-dimensional spaces (we provide the formal definition for this modification in Appendix D.1). The left panel of Figure 2 shows that the loss landscape of $\boldsymbol{w}$ forms two basins at the locations of the ID tokens. The central panel shows that after training, $\boldsymbol{w}$ is located in one of the basins. Finally, the right panel shows that AP-OOD perfectly discriminates ID and OOD.

**Summarization.** We follow Ren et al. (2023) and use a PEGASUS$_{\text{LARGE}}$ (Zhang et al., 2020) fine-tuned on the ID data set XSUM (Narayan et al., 2018). We utilize the C4 training split as the AUX data set. We measure the OOD detection performance on the data sets CNN/Daily Mail (CNN/DM; news articles from CNN and Daily Mail; Hermann et al., 2015; See et al., 2017), Newsroom (articles and summaries written by authors and editors from 38 news publications; Grusky et al., 2018), Reddit TIFU (posts and summaries from the online discussion forum Reddit; Kim et al., 2018), and Samsum (summaries of casual dialogues; Gliwa et al., 2019). The ForumSum data set used in the experiments of Ren et al. (2023) has been retracted. Therefore, we do not use it in our experiments.

**Translation.** We train a Transformer (base) on WMT15 En–Fr (Bojar et al., 2015). The model trains for 100,000 steps using AdamW (Loshchilov & Hutter, 2017) with a cosine schedule (Loshchilov & Hutter, 2016), linear warmup, and a peak learning rate of $5 \times 10^{-4}$.

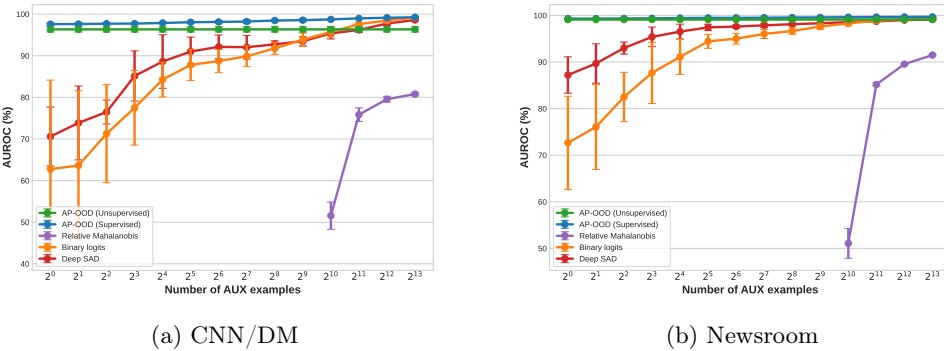

(a) CNN/DM          (b) Newsroom

Figure 3: OOD detection performance on the input token embeddings of PEGASUS$_{\text{LARGE}}$ trained on XSUM. We vary the number of AUX samples and compare AP-OOD, binary logits (Ren et al., 2023), Deep SAD (Ruff et al., 2019), and relative Mahalanobis (Ren et al., 2023). AP-OOD attains the highest AUROC independent of AUX sample count.

We set the batch size to 1024 and the context length to 512. Following Ren et al. (2023), the AUX data set is ParaCrawl En–Fr, and the OOD data sets are newstest2014 (nt2014), newsdiscussdev2015 (ndd2015), and newsdiscusstest2015 (ndt2015) from WMT15 (Bojar et al., 2015), and the Law, Koran, Medical, IT, and Subtitles subsets from OPUS (Tiedemann, 2012; Aulamo & Tiedemann, 2019).

**Training.** We extract 100,000 ID sequence representations ($\boldsymbol{E}$ or $\boldsymbol{G}$) and use all extracted representations for training AP-OOD in all experiments. We also extract AUX sequence representations, and we vary the number of AUX sequences available from 0 (unsupervised) to 10,000 (fully supervised). While training AP-OOD, the transformer model remains frozen. We use the Adam optimizer (Kingma & Ba, 2014) without weight decay, set the learning rate to 0.01, and apply a cosine schedule (Loshchilov & Hutter, 2016). We train for 1,000 steps with a batch size of 512. We select $M$ and $T$ such that the parameter count of AP-OOD matches the parameter count of the Mahalanobis method (i.e., the size of $\boldsymbol{\Sigma}$). For more information on hyperparameter selection, we refer to Appendix D.2. An additional scaling experiment on input sequence representations of the summarization task investigates larger parameter spaces: We train on the full XSUM data set (instead of the 100,000 ID sequence representations used in the other experiments). We select the number of heads ($M$) from the set $\{1, 16, 128, 1024\}$, the number of queries ($T$) from the set $\{1, 4, 16\}$, and $\beta$ from $\{1/\sqrt{D}, 0.25, 0.5, 1, 2\}$. The largest configuration has a parameter count 16 times greater than the Mahalanobis baseline.

**Baselines.** We compare AP-OOD to six unsupervised OOD detection methods: We apply the embedding-based methods Mahalanobis (Lee et al., 2018; Ren et al., 2023), KNN (Sun et al., 2022), and Deep SVDD (Ruff et al., 2018) to both the input and output sequence representations ($\boldsymbol{E}$ and $\boldsymbol{G}$, respectively), and we apply Perplexity (Ren et al., 2023) and Entropy (Malinin & Gales, 2020) to the output of the decoder. We also compare AP-OOD to three supervised OOD detection methods: binary logits (Ren et al., 2023), relative Mahalanobis (Ren et al., 2023), and Deep SAD (Ruff et al., 2019). We evaluate the discriminative power of the methods in our comparison using the false positive rate at 95% true positives (FPR95) and AUROC.

**Audio data.** To demonstrate the effectiveness of AP-OOD on data modalities other than text, we apply the method to the MIMII-DG audio data set (Dohi et al., 2022). The data set comprises audio recordings of 15 different machines, ranging from 10 to 12 seconds in length. The dataset contains 990 samples per machine. During preprocessing, the raw audio waveforms are converted into audio spectrograms. We train a transformer to classify a subset of 7 machines. The remaining 8 machines are considered as OOD. The architecture and training method for the network were adopted from Huang et al. (2022). To adjust for the

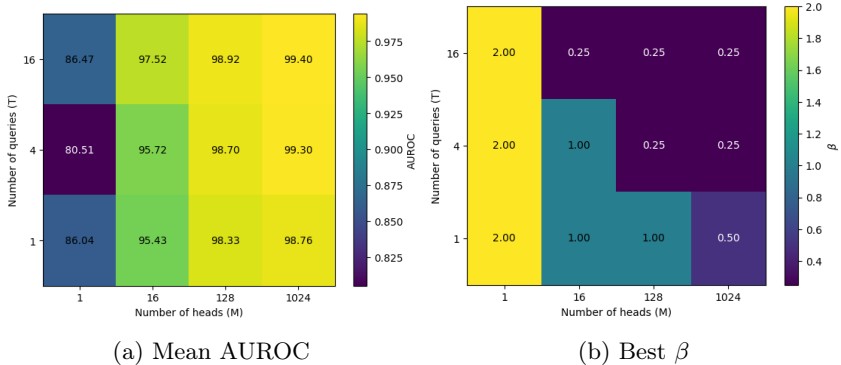

(a) Mean AUROC                                  (b) Best $\beta$

Figure 4: OOD detection performance on the input token embeddings of $\text{PEGASUS}_{\text{LARGE}}$ trained on XSUM when scaling to large $M$ and $T$. We vary $M \in \{1, 16, 128, 1024\}$, $T \in \{1, 4, 16\}$, and $\beta \in \{1/\sqrt{D}, 0.25, 0.5, 1, 2\}$. **(Left)** Mean AUROC in % for the best $\beta$ at each $(M, T)$ combination. **(Right)** The best $\beta$ selected at each $(M, T)$ combination.

small data set size, we decrease the size of the architecture: We increase the patch size to $32 \times 32$ pixels, decrease the embedding dimension to 32, and utilize only three attention blocks with four heads each. Consequently, the encoder of the network produces 128 tokens with $D = 32$ features. We train AP-OOD on the encoder output in the unsupervised setting using $M = 128$ heads and $T = 8$ queries.

## 4 RESULTS & DISCUSSION

Table 1 shows the results on unsupervised OOD detection on text summarization. AP-OOD surpasses methods with mean pooling by a large margin for both input and output settings for most OOD data sets. Most notably, the mean FPR95 in the input setting on CNN/DM improves from 74.20% for the best baseline Deep SVDD to 12.88% for AP-OOD. The table also shows that the embedding-based methods (Mahalanobis, KNN, Deep SVDD, and AP-OOD) perform better than the prediction-based baselines perplexity and entropy. Figure 3 shows the results of AP-OOD in the semi-supervised setting: supplying AUX data to AP-OOD improves the AUROC, and more AUX data results in a larger improvement. AP-OOD attains the highest AUROC independent of AUX sample count. We include the results on additional OOD data sets in the semi-supervised setting and results on fully supervised OOD detection on text summarization in Appendix D.3, and we present ablations on AP-OOD on text summarization in Appendix D.8. Furthermore, we verify the effectiveness of AP-OOD on the decoder-only language modeling paradigm using Pythia-160M in Appendix D.6.

Figure 4 presents the results when scaling AP-OOD to larger parameter counts. As we increase the number of heads ($M$) and queries ($T$), we observe a steady increase in the mean AUROC on the summarization task. The highest mean AUROC of 99.40% is achieved by the largest configuration tested ($M = 1024$, $T = 16$).

Table 2 shows the results on unsupervised OOD detection on the translation task. AP-OOD gives the best average results for the input and output settings. We include results on fully supervised OOD detection for translation in Appendix D.5. Unlike in the summarization task, the prediction-based methods (perplexity and entropy) are competitive with embedding-based methods in the translation task. Notably, perplexity outperforms all embedding-based methods evaluated on output token embeddings, with the exception of AP-OOD. We explain this discrepancy through the varying levels of aleatoric uncertainty inherent in the two tasks. In translation, the source sentence imposes strict lexical and syntactic constraints, resulting in low-entropy output distributions and therefore low aleatoric uncertainty. In contrast, summarization yields multiple distinct yet equally valid outputs, thereby increasing output entropy and, consequently, the aleatoric uncertainty. Prediction-based uncertainty estimators are inherently limited in settings with non-trivial aleatoric uncertainty (Tomov et al., 2025).

Table 2: Unsupervised OOD detection performance on English-to-French translation. We compare results from AP-OOD, Mahalanobis (Lee et al., 2018; Ren et al., 2023), KNN (Sun et al., 2022), Deep SVDD (Ruff et al., 2018), model perplexity (Ren et al., 2023), and entropy (Malinin & Gales, 2020) on a Transformer (base) trained on WMT15 En–Fr as the ID data set. ↓ indicates "lower is better" and ↑ "higher is better". All values in %. We estimate standard deviations across five independent data set splits and training runs.

| | | IT | Koran | Law | Medical | Subtitles | ndd2015 | ndt2015 | nt2014 | Mean |
|---|---|---|---|---|---|---|---|---|---|---|
| | | | | | Input OOD | | | | | |
| Mahalanobis | AUROC ↑ | $93.28^{\pm0.03}$ | $\underline{74.52^{\pm0.37}}$ | $60.48^{\pm0.51}$ | $\underline{78.63^{\pm0.25}}$ | $87.27^{\pm0.13}$ | $62.02^{\pm0.04}$ | $62.40^{\pm0.02}$ | $49.08^{\pm0.05}$ | 70.96 |
| | FPR95 ↓ | $37.80^{\pm0.13}$ | $\underline{93.58^{\pm0.17}}$ | $90.79^{\pm0.29}$ | $\underline{67.14^{\pm0.47}}$ | $\underline{70.73^{\pm0.48}}$ | $91.67^{\pm0.09}$ | $93.60^{\pm0.11}$ | $98.59^{\pm0.03}$ | 80.49 |
| KNN | AUROC ↑ | $\underline{94.25^{\pm0.01}}$ | $71.52^{\pm0.20}$ | $54.79^{\pm0.20}$ | $77.86^{\pm0.23}$ | $\mathbf{87.47^{\pm0.10}}$ | $\underline{63.66^{\pm0.11}}$ | $65.16^{\pm0.11}$ | $\underline{53.27^{\pm0.07}}$ | 71.00 |
| | FPR95 ↓ | $\underline{35.00^{\pm0.17}}$ | $93.91^{\pm0.17}$ | $91.73^{\pm0.26}$ | $68.95^{\pm0.18}$ | $71.91^{\pm0.43}$ | $\underline{91.33^{\pm0.08}}$ | $92.07^{\pm0.08}$ | $\underline{97.28^{\pm0.02}}$ | 80.27 |
| Deep SVDD | AUROC ↑ | $92.15^{\pm0.17}$ | $72.26^{\pm0.60}$ | $\mathbf{63.57^{\pm3.22}}$ | $78.46^{\pm0.48}$ | $85.06^{\pm0.48}$ | $59.59^{\pm0.39}$ | $60.00^{\pm0.23}$ | $46.70^{\pm0.12}$ | 69.72 |
| | FPR95 ↓ | $46.28^{\pm0.86}$ | $96.11^{\pm0.46}$ | $91.53^{\pm1.02}$ | $68.94^{\pm0.94}$ | $74.64^{\pm1.84}$ | $93.77^{\pm0.28}$ | $94.84^{\pm0.53}$ | $98.82^{\pm0.10}$ | 83.12 |
| AP-OOD (Ours) | AUROC ↑ | $\mathbf{95.61^{\pm0.09}}$ | $\mathbf{78.49^{\pm1.65}}$ | $\underline{63.23^{\pm3.77}}$ | $\mathbf{80.16^{\pm0.58}}$ | $\underline{87.39^{\pm1.37}}$ | $\mathbf{67.64^{\pm1.40}}$ | $\mathbf{69.50^{\pm1.41}}$ | $\mathbf{56.44^{\pm1.13}}$ | **74.81** |
| | FPR95 ↓ | $\mathbf{21.85^{\pm1.25}}$ | $91.07^{\pm1.38}$ | $89.19^{\pm0.39}$ | $\mathbf{61.55^{\pm1.08}}$ | $66.35^{\pm3.31}$ | $87.85^{\pm1.14}$ | $89.07^{\pm0.96}$ | $95.10^{\pm0.17}$ | **75.25** |
| | | | | | Output OOD | | | | | |
| Perplexity | AUROC ↑ | $91.34^{\pm0.00}$ | $71.75^{\pm0.15}$ | $45.13^{\pm0.22}$ | $73.05^{\pm0.35}$ | $\underline{91.27^{\pm0.15}}$ | $72.80^{\pm0.00}$ | $\underline{73.43^{\pm0.00}}$ | $59.21^{\pm0.00}$ | 72.25 |
| | FPR95 ↓ | $44.94^{\pm0.00}$ | $91.70^{\pm0.45}$ | $90.83^{\pm0.12}$ | $70.47^{\pm0.36}$ | $\underline{51.31^{\pm0.79}}$ | $85.05^{\pm0.05}$ | $85.67^{\pm0.00}$ | $96.67^{\pm0.00}$ | $\underline{77.08}$ |
| Entropy | AUROC ↑ | $74.86^{\pm0.20}$ | $\mathbf{85.96^{\pm0.32}}$ | $53.38^{\pm0.47}$ | $50.51^{\pm0.35}$ | $70.36^{\pm0.23}$ | $\underline{73.78^{\pm0.21}}$ | $72.57^{\pm0.10}$ | $\mathbf{71.01^{\pm0.37}}$ | 69.05 |
| | FPR95 ↓ | $68.69^{\pm0.91}$ | $\mathbf{55.03^{\pm1.18}}$ | $93.76^{\pm0.15}$ | $90.21^{\pm0.24}$ | $75.84^{\pm1.19}$ | $\mathbf{76.64^{\pm1.17}}$ | $\mathbf{77.63^{\pm1.31}}$ | $\mathbf{85.39^{\pm0.79}}$ | 77.90 |
| Mahalanobis | AUROC ↑ | $91.07^{\pm0.03}$ | $75.57^{\pm0.39}$ | $64.15^{\pm0.72}$ | $78.22^{\pm0.18}$ | $86.68^{\pm0.07}$ | $62.76^{\pm0.05}$ | $63.53^{\pm0.06}$ | $49.70^{\pm0.09}$ | 71.46 |
| | FPR95 ↓ | $54.49^{\pm0.23}$ | $93.82^{\pm0.14}$ | $92.31^{\pm0.17}$ | $73.03^{\pm0.39}$ | $74.97^{\pm0.46}$ | $92.87^{\pm0.15}$ | $93.97^{\pm0.04}$ | $98.82^{\pm0.03}$ | 84.28 |
| KNN | AUROC ↑ | $\mathbf{95.05^{\pm0.01}}$ | $75.31^{\pm0.22}$ | $\mathbf{65.01^{\pm0.19}}$ | $\mathbf{82.39^{\pm0.11}}$ | $85.87^{\pm0.11}$ | $65.78^{\pm0.16}$ | $66.72^{\pm0.16}$ | $58.99^{\pm0.12}$ | $\underline{74.39}$ |
| | FPR95 ↓ | $\underline{31.95^{\pm0.12}}$ | $95.56^{\pm0.20}$ | $91.88^{\pm0.16}$ | $\underline{65.91^{\pm0.46}}$ | $79.04^{\pm0.23}$ | $92.91^{\pm0.00}$ | $93.99^{\pm0.07}$ | $97.54^{\pm0.02}$ | 81.10 |
| Deep SVDD | AUROC ↑ | $89.57^{\pm0.26}$ | $70.96^{\pm1.27}$ | $\underline{64.36^{\pm1.37}}$ | $76.47^{\pm0.27}$ | $82.52^{\pm0.51}$ | $60.39^{\pm0.42}$ | $61.48^{\pm0.34}$ | $48.03^{\pm0.60}$ | 69.22 |
| | FPR95 ↓ | $57.89^{\pm1.03}$ | $93.97^{\pm0.44}$ | $91.34^{\pm0.52}$ | $72.80^{\pm1.09}$ | $82.77^{\pm0.91}$ | $94.73^{\pm0.19}$ | $93.91^{\pm0.36}$ | $98.53^{\pm0.24}$ | 85.74 |
| AP-OOD (Ours) | AUROC ↑ | $94.44^{\pm0.53}$ | $\underline{81.50^{\pm1.51}}$ | $57.36^{\pm1.70}$ | $\underline{78.74^{\pm1.26}}$ | $\mathbf{91.39^{\pm1.41}}$ | $\mathbf{75.65^{\pm0.96}}$ | $\mathbf{75.41^{\pm1.01}}$ | $\underline{64.55^{\pm0.99}}$ | **77.38** |
| | FPR95 ↓ | $\mathbf{26.42^{\pm1.37}}$ | $\underline{80.45^{\pm2.58}}$ | $88.24^{\pm0.85}$ | $62.26^{\pm1.18}$ | $49.36^{\pm7.09}$ | $\underline{80.44^{\pm2.35}}$ | $81.56^{\pm2.18}$ | $94.22^{\pm0.82}$ | **70.37** |

Table 3: Unsupervised OOD detection performance on audio classification. We compare results from AP-OOD, Mahalanobis (Lee et al., 2018; Ren et al., 2023), KNN (Sun et al., 2022), Deep SVDD (Ruff et al., 2018), MSP (Hendrycks & Gimpel, 2016), and EBO (Liu et al., 2020b) trained on MIMII-DG (Dohi et al., 2022) as the ID data set. ↓ indicates "lower is better" and ↑ "higher is better". All values in %. We estimate standard deviations across five independent training runs.

| | Mahalanobis | KNN | Deep SVDD | MSP | EBO | AP-OOD (Ours) |
|---|---|---|---|---|---|---|
| AUROC ↑ | $64.96^{\pm0.002}$ | $81.21^{\pm0.000}$ | $53.48^{\pm1.930}$ | $88.05^{\pm0.000}$ | $90.75^{\pm0.000}$ | $\mathbf{92.86^{\pm0.746}}$ |
| FPR95 ↓ | $84.39^{\pm0.011}$ | $57.11^{\pm0.000}$ | $89.44^{\pm1.689}$ | $36.43^{\pm0.000}$ | $61.86^{\pm0.000}$ | $\mathbf{22.35^{\pm2.388}}$ |

In the audio task, the network achieves an accuracy of 97.6% on the primary classification task. Table 3 presents the results of the unsupervised OOD detection methods AP-OOD, Mahalanobis (Lee et al., 2018), KNN (Sun et al., 2022), and Deep SVDD (Ruff et al., 2018). Additionally, we compare AP-OOD to 2 methods for classifiers, Maximum Softmax Probability (MSP; Hendrycks & Gimpel, 2016) and Energy-based OOD Detection (EBO; Liu et al., 2020b). In contrast to the other methods, MSP and EBO do not apply to transformer tokens, making them unsuitable for summarization and translation tasks. The results show that AP-OOD improves the FPR95 metric from 36.43% (MSP) to 22.35%.

We evaluate the runtime performance of AP-OOD by measuring the inference time of single batches on the summarization task. We find that while AP-OOD is slower than the Mahalanobis baseline, it is still substantially faster than a forward pass through the transformer encoder. Because AP-OOD scales linearly, its relative overhead diminishes for longer sequences. For more details on the runtime behavior, we refer to Appendix D.9.

## 5 Related Work

**OOD detection.** Some authors (e.g., Bishop, 1994; Roth et al., 2022; Yang et al., 2022) distinguish between anomalies, outliers, and novelties. These distinctions reflect different goals within applications (Ruff et al., 2021). For example, when an anomaly is found, it will usually be removed from the training pipeline. However, when a novelty is found, it should be studied. We focus on detecting samples that are not part of the training distribution and consider sample categorization as a downstream task. OOD detection methods can be categorized into three groups: Post-hoc, training-time, and OE methods. In the post-hoc

approach, one employs statistics obtained from a classifier. Perhaps the most well-known approach is the maximum softmax probability (MSP; Hendrycks & Gimpel, 2016). A wide range of post-hoc OOD detection approaches have been proposed to address the shortcomings of MSP (e.g., Lee et al., 2018; Hendrycks et al., 2019a; Liu et al., 2020a; Sun et al., 2021; 2022; Wang et al., 2022; Zhang et al., 2023b; Djurisic et al., 2023; Liu et al., 2023; Xu et al., 2024; Guo et al., 2025). A commonly used post-hoc method is the Mahalanobis distance (e.g., Lee et al., 2018; Sehwag et al., 2021; Ren et al., 2023). Recently, Müller & Hein (2025) proposed feature normalization to improve Mahalanobis-based OOD detection, and Guo et al. (2025) show that the Mahalanobis distance benefits from dynamically adjusting the prior geometry in response to new data. In contrast to post-hoc methods, training-time methods modify the training process of the encoder (e.g., Hendrycks et al., 2019c; Sehwag et al., 2021; Du et al., 2022; Hendrycks et al., 2022; Ming et al., 2023; Tao et al., 2023; Lu et al., 2024). Finally, the group of OE methods incorporates AUX data in the training process (e.g., Hendrycks et al., 2019b; Liu et al., 2020a; Ming et al., 2022; Zhang et al., 2023a; Wang et al., 2023; Zhu et al., 2023; Jiang et al., 2024; Hofmann et al., 2024).

**OOD detection and natural language.** Most of the aforementioned OOD detection approaches target vision tasks, and many of them require a classification model as the encoder $\phi$. Applying these vision-based OOD methods to text is not straightforward due to the sequence-dependent nature of natural language (e.g., in autoregressive language generation). OOD detection specifically tailored for natural language is still underexplored. Ren et al. (2023) propose the model's log-perplexity of a generated sequence $\boldsymbol{y}$ as a simple baseline for OOD detection on conditional language modeling tasks: $-\frac{1}{L}\sum_{l=1}^{L}\log p_\theta(y_l|\boldsymbol{y}_{<l},\boldsymbol{x})$. However, they show experimentally that model perplexity is inherently limited. Because of these shortcomings, Ren et al. (2023) propose embedding-based OOD detection methods for text data. Relatively few other works have explored OOD detection for generative language modeling. Notable applications include translation (e.g., Xiao et al., 2020; Malinin et al., 2021; Ren et al., 2023), summarization (Ren et al., 2023), and mathematical reasoning (Wang et al., 2024). A related field is hallucination detection (e.g., Malinin & Gales, 2020; Farquhar et al., 2024; Du et al., 2024; Aichberger et al., 2025; Park et al., 2025). Unlike OOD detection (which flags inputs outside the training distribution), the goal of many hallucination detection methods is to identify prompts a generative language model is unlikely to answer truthfully.

## 6 Limitations & Future Work

We would like to discuss two limitations that we found. First, the selection of the AUX data is crucial, since it determines the shape of the ID–OOD decision boundary. If the AUX distribution diverges from the OOD examples faced at inference, the induced boundary may not be aligned with the task. Second, it remains unclear how reliably the OOD detection performance on specific data sets can indicate the general ability to detect OOD examples, as a large portion of plausible OOD inputs remains untested. An interesting avenue for future work is to apply OOD detection methods to large language models (LLMs; e.g., Abdin et al., 2024; Dubey et al., 2024; Yang et al., 2025). While we demonstrate the applicability of AP-OOD on the decoder-only language modeling paradigm of LLMs (Appendix D.6), further challenges include proprietary training data, finding OOD data for training and evaluation given the breadth of the ID data, and the ambiguity of $p_{\text{ID}}$ arising from complex training pipelines involving multiple phases (e.g., Wei et al., 2022; Ouyang et al., 2022).

## 7 Conclusion

We introduce AP-OOD: an approach for OOD detection for natural language that can learn in supervised and unsupervised settings. In contrast to previous methods, AP-OOD learns how to pool token-level information without the explicit need for AUX data. Our experiments show that when supplied with AUX data during training, the performance of AP-OOD improves as more AUX data is provided. We compare AP-OOD to five unsupervised and three supervised OOD detection methods. Overall, AP-OOD shows the best results.

ACHKNOWLEDGEMENTS

The ELLIS Unit Linz, the LIT AI Lab, the Institute for Machine Learning, are supported by the Federal State Upper Austria. We thank the projects FWF AIRI FG 9-N (10.55776/FG9), AI4GreenHeatingGrids (FFG- 899943), Stars4Waters (HORIZON-CL6-2021-CLIMATE-01-01), FWF Bilateral Artificial Intelligence (10.55776/COE12). We thank NXAI GmbH, Audi AG, Silicon Austria Labs (SAL), Merck Healthcare KGaA, GLS (Univ. Waterloo), TÜV Holding GmbH, Software Competence Center Hagenberg GmbH, dSPACE GmbH, TRUMPF SE + Co. KG. This work has been supported by the "University SAL Labs" initiative of Silicon Austria Labs (SAL) and its Austrian partner universities for applied fundamental research for electronic-based systems. We acknowledge EuroHPC Joint Undertaking for awarding us access to Karolina at IT4Innovations, Czech Republic and Leonardo at CINECA, Italy.

REPRODUCIBILITY STATEMENT

To ensure reproducibility, the source code of our implementation of AP-OOD in the unsupervised and supervised settings is available at https://github.com/ml-jku/ap-ood. We provide information about data, the training process, and the hyperparameter selection in Section 3 and Appendix D.2.

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

TABLE OF CONTENTS

## A    RELATED WORK

**Continuous modern Hopfield networks.**    Modern Hopfield networks (MHNs) are energy-based associative memory networks. They advance conventional Hopfield networks (Hopfield, 1984) by introducing continuous queries and states and a new energy function. MHNs have exponential storage capacity, while retrieval is possible with a one-step update (Ramsauer et al., 2021). The update rule of MHNs coincides with attention as it is used in the Transformer (Vaswani et al., 2017a). Examples for successful applications of MHNs are Widrich et al. (2020a); Fürst et al. (2022); Sanchez-Fernandez et al. (2022); Paischer et al. (2022); Schäfl et al. (2022); Schimunek et al. (2023); Auer et al. (2023) and Hofmann et al. (2024).

**Multiple instance learning (MIL).**    MIL (Dietterich et al., 1997; Maron & Lozano-Pérez, 1997; Andrews et al., 2002; Ilse et al., 2018) considers a classifier that maps a bag $\boldsymbol{Z} = (\boldsymbol{z}_1, \ldots, \boldsymbol{z}_S)$ of instances $\boldsymbol{z}_s$ to a bag-level label $Y \in \{0, 1\}$. MIL also assumes that individual labels $y_s \in \{0, 1\}$ exist for the instances, which remain unknown during training. By assumption, the bag-level label is positive once one of the instance-level labels is positive (and negative if all are instance-level labels negative), i.e., $Y := \max_s y_s$. Recent MIL methods use attention pooling (Ilse et al., 2018; Shao et al., 2021; Al Hajj et al., 2024) and modern Hopfield networks (Widrich et al., 2020b) to pool the features of the instances.

**One-class classification (OCC).**    OCC (Schölkopf et al., 1999) is the problem of learning a decision boundary separating the ID and OOD regions while having access to examples from the ID data set only. One-Class SVM (Schölkopf et al., 2001) learns a maximum margin hyperplane in the feature space that separates the ID data from the origin. Support Vector Data Description (SVDD; Tax & Duin, 2004) learns a hypersphere which encapsulates the ID data. Most closely related to AP-OOD is Deep SVDD (Ruff et al., 2018). Deep SVDD learns an encoder $\psi(\cdot, \mathcal{W}) : \mathbb{R}^D \to \mathbb{R}^M$ by minimizing the volume of a data-enclosing hypersphere in the output space. Ruff et al. (2019) propose Deep SAD, an extension of Deep SVDD that makes use of AUX data during training. However, Liznerski et al. (2022) show that the effectiveness of this extension degrades with increasing dimensionality.

## B    Theoretical Notes

### B.1    OOD Score Investigation

In the following, we show that

$$\min_{j \in \{1, \ldots, M\}} -d_j^2(\phi_{\text{enc}}(\boldsymbol{x}), \tilde{\boldsymbol{Z}}) + \log(||\boldsymbol{w}_j||_2^2) < 2\log(\epsilon) \ + \ \log(2\pi) \quad \Longrightarrow \quad \boldsymbol{x} \in \mathbb{O}$$

whenever $z_j := \frac{\boldsymbol{w}_j^T}{||\boldsymbol{w}_j||_2} \bar{\boldsymbol{z}}_j$ is normally distributed with probability density function

$$\dot{p}_j(z_j) \ := \ \frac{||\boldsymbol{w}_j||_2}{\sqrt{2\pi}} \ \exp\left( -\frac{1}{2}(||\boldsymbol{w}_j||_2 \, z_j \ - \ \boldsymbol{w}_j^T \boldsymbol{\mu}_j)^2 \right), \tag{16}$$

weight vectors $\boldsymbol{w}_j \in \mathbb{R}^D$, encoder $\phi_{\text{enc}} : \mathcal{X} \to \mathcal{Z}$, $\mathcal{Z} = \bigcup_{S \geq 1} \mathbb{R}^{D \times S}$, $\boldsymbol{Z} \in \mathcal{Z}$, $\tilde{\boldsymbol{Z}} \in \mathcal{Z}$, $\bar{\boldsymbol{z}}_j = \boldsymbol{Z} \boldsymbol{p}_j$, $\boldsymbol{\mu}_j = \tilde{\boldsymbol{Z}} \tilde{\boldsymbol{p}}_j$ , $\boldsymbol{p}_j \in \Delta^{S-1}$ and $\tilde{\boldsymbol{p}}_j \in \Delta^{S'-1}$ with

$$\Delta^{S-1} := \Big\{ (p_1, \ldots, p_S) \in [0,1]^S \ | \ \sum_{i=1}^{S} p_i = 1 \Big\}.$$

*Proof.* Note that the $\phi_{\text{enc}}$-pushforward density $p_{\phi_{\text{enc}}}$ of $p_{\text{ID}}$ satisfies

$$p_{\phi_{\text{enc}}}(\boldsymbol{Z}) := \sum_{\boldsymbol{x} \in \mathcal{X}} p_{\text{ID}}(\boldsymbol{x}) \, \delta(\phi_{\text{enc}}(\boldsymbol{x}) = \boldsymbol{Z}) \ \geq \ p_{\text{ID}}(\boldsymbol{x}).$$

Analogously, we get $\bar{p}_j(\bar{\boldsymbol{z}}_j) \geq p_{\phi_{\text{enc}}}(\boldsymbol{Z})$ for $\bar{\boldsymbol{z}}_j := \boldsymbol{Z} \boldsymbol{p}_j$ and $\dot{p}_j(z_j) \geq \bar{p}_j(\bar{\boldsymbol{z}}_j)$ for $z_j := \frac{\boldsymbol{w}_j^T}{||\boldsymbol{w}_j||_2} \bar{\boldsymbol{z}}_j$.

That is, for any $j \in \{1, \ldots, M\}$, we have that $p_{\text{ID}}(\boldsymbol{x}) \leq p_{\phi_{\text{enc}}}(\boldsymbol{Z}) \leq \bar{p}_j(\bar{\boldsymbol{z}}_j) \leq \dot{p}_j(z_j)$. As a consequence, for all $j \in \{1, \ldots, M\}$ it holds that $\dot{p}_j(z_j) < \epsilon \implies p_{\text{ID}}(\boldsymbol{x}) < \epsilon$. Moreover, the following equivalence holds:

$$\dot{p}_j(z_j) \ < \ \epsilon \qquad\qquad \Longleftrightarrow$$

$$\frac{||\boldsymbol{w}_j||_2}{\sqrt{2\pi}} \ \exp\left( -\frac{1}{2}(||\boldsymbol{w}_j||_2 \, z_j \ - \ \boldsymbol{w}_j^T \boldsymbol{\mu}_j)^2 \right) \ < \ \epsilon \qquad\qquad \Longleftrightarrow$$

$$\frac{||\boldsymbol{w}_j||_2}{\sqrt{2\pi}} \exp\left( -\frac{1}{2}(\boldsymbol{w}_j^T \bar{\boldsymbol{z}}_j \ - \ \boldsymbol{w}_j^T \boldsymbol{\mu}_j)^2 \right) \ < \ \epsilon \qquad\qquad \Longleftrightarrow$$

$$- (\boldsymbol{w}_j^T \bar{\boldsymbol{z}}_j \ - \ \boldsymbol{w}_j^T \boldsymbol{\mu}_j)^2 \ + \ \log(||\boldsymbol{w}_j||_2^2) \ < \ 2\log(\epsilon) \ + \ \log(2\pi) \tag{17}$$

As a consequence, we have that $\boldsymbol{x} \in \mathbb{O}$, if Equation (17) is satisfied for any $j \in \{1, \ldots, M\}$.    $\square$

### B.2    Mahalanobis Decomposition

We assume the $D$ weight vectors $\boldsymbol{w}_j$ are linearly independent. First, we start from the directional decomposition and show the relation to the Mahalanobis distance.

$$d_{\text{Maha}}^2(\bar{\boldsymbol{z}}, \boldsymbol{\mu}) \ = \ \sum_{j=1}^{D} \left( \boldsymbol{w}_j^T \bar{\boldsymbol{z}} - \ \boldsymbol{w}_j^T \boldsymbol{\mu} \right)^2 \tag{18}$$

$$= (\bar{\boldsymbol{z}} \ - \ \boldsymbol{\mu})^T \left( \sum_{i=1}^{D} \boldsymbol{w}_j \boldsymbol{w}_j^T \right) (\bar{\boldsymbol{z}} \ - \ \boldsymbol{\mu}) \tag{19}$$

$$= (\bar{\boldsymbol{z}} \ - \ \boldsymbol{\mu})^T \boldsymbol{\Sigma}^{-1} (\bar{\boldsymbol{z}} \ - \ \boldsymbol{\mu}). \tag{20}$$

Because the weight vectors are linearly independent, $\boldsymbol{\Sigma}^{-1}$ has full rank. Next, we go in the opposite direction and show that the eigenvectors $\boldsymbol{V} = (\boldsymbol{v}_1, \ldots, \boldsymbol{v}_D)$ and eigenvalues $\boldsymbol{D} = \text{diag}(\lambda_1, \ldots, \lambda_D)$ of a $\boldsymbol{\Sigma}$ can be used to define corresponding $\boldsymbol{w}_j$.

$$d^2_{\text{Maha}}(\bar{z}, \boldsymbol{\mu}) = (\bar{z} - \boldsymbol{\mu})^T \boldsymbol{\Sigma}^{-1} (\bar{z} - \boldsymbol{\mu}) \tag{21}$$

$$= (\bar{z} - \boldsymbol{\mu})^T \boldsymbol{V}^T \boldsymbol{D}^{-1} \boldsymbol{V} (\bar{z} - \boldsymbol{\mu}) \tag{22}$$

$$= \left( \sqrt{\boldsymbol{D}^{-1}} \boldsymbol{V} \bar{z} - \sqrt{\boldsymbol{D}^{-1}} \boldsymbol{V} \boldsymbol{\mu} \right)^T \left( \sqrt{\boldsymbol{D}^{-1}} \boldsymbol{V} \bar{z} - \sqrt{\boldsymbol{D}^{-1}} \boldsymbol{V} \boldsymbol{\mu} \right) \tag{23}$$

$$= \sum_{j=1}^{D} (\boldsymbol{w}_j^T \bar{z} - \boldsymbol{w}_j^T \boldsymbol{\mu})^2, \tag{24}$$

where $\boldsymbol{w}_j = \sqrt{\lambda_j^{-1}} \boldsymbol{v}_j$, $\boldsymbol{\Sigma} = \boldsymbol{V}^T \boldsymbol{D} \boldsymbol{V}$, and $\boldsymbol{\Sigma}^{-1} = \boldsymbol{V}^T \boldsymbol{D}^{-1} \boldsymbol{V}$.

The relation between the Mahalanobis distance and the directional decomposition is as follows:

*1. Any linearly independent sequence $\boldsymbol{w}_1, \ldots, \boldsymbol{w}_D$ induces a positive definite matrix $\boldsymbol{\Sigma}^{-1} := \sum_{j=1}^{D} \boldsymbol{w}_j \boldsymbol{w}_j^\top$, and hence a Mahalanobis distance satisfying*

$$\sum_{j=1}^{D} (\boldsymbol{w}_j^\top \bar{z} - \boldsymbol{w}_j^\top \boldsymbol{\mu})^2 = (\bar{z} - \boldsymbol{\mu})^\top \boldsymbol{\Sigma}^{-1} (\bar{z} - \boldsymbol{\mu}). \tag{25}$$

*2. Conversely, any full-rank covariance matrix $\boldsymbol{\Sigma}$ admits a set of linearly independent vectors $\boldsymbol{w}_1, \ldots, \boldsymbol{w}_D$ such that $\boldsymbol{\Sigma}^{-1} = \sum_{j=1}^{D} \boldsymbol{w}_j \boldsymbol{w}_j^\top$, and therefore Equation (25) holds.*

Thus, our decomposition and the Mahalanobis form represent the same quadratic form; the eigen-decomposition is only one possible choice of $\boldsymbol{w}_j$.

### B.3 AP-OOD: KERNEL VIEW

In this section, we express the distance function $d^2$ using kernels to gain further insight into its properties. We use $d^2$ as defined in Equation (13):

$$d^2(\boldsymbol{Z}, \tilde{\boldsymbol{Z}}) := \sum_{j=1}^{M} \left( \text{Tr}(\boldsymbol{W}_j^T \boldsymbol{Z} \text{softmax}(\beta \, \boldsymbol{Z}^T \boldsymbol{W}_j)) - \text{Tr}(\boldsymbol{W}_j^T \tilde{\boldsymbol{Z}} \text{softmax}(\beta \, \tilde{\boldsymbol{Z}}^T \boldsymbol{W}_j)) \right)^2, \tag{26}$$

where $\boldsymbol{Z}, \tilde{\boldsymbol{Z}} \in \mathcal{Z}$, are the sequence representation and the concatenated sequence representation, respectively, $\boldsymbol{W}_j \in \mathbb{R}^{D \times T}$, are the weights of head $j$ for $j \in \{1, \ldots, M\}$, and $\beta \in \mathbb{R}_{\geq 0}$ is the inverse temperature. Recall that $\boldsymbol{Z} = (\boldsymbol{z}_1, \ldots, \boldsymbol{z}_S)$ and $\boldsymbol{W}_j = (\boldsymbol{w}_{j1}, \ldots, \boldsymbol{w}_{jT})$.

For simplicity, we denote the output of $d^2(\boldsymbol{Z}, \tilde{\boldsymbol{Z}})$ as $d^2$ for given instantiations of $\boldsymbol{Z}, \tilde{\boldsymbol{Z}}$. We start by writing $d^2$ as

$$d^2 = \sum_{j=1}^{M} (s_j - \mu_j)^2, \tag{27}$$

$$s_j := \text{Tr}(\boldsymbol{W}_j^T \boldsymbol{Z} \text{softmax}(\beta \, \boldsymbol{Z}^T \boldsymbol{W}_j)), \tag{28}$$

$$\mu_j := \text{Tr}(\boldsymbol{W}_j^T \tilde{\boldsymbol{Z}} \text{softmax}(\beta \, \tilde{\boldsymbol{Z}}^T \boldsymbol{W}_j)). \tag{29}$$

We first investigate $s_j$:

$$s_j := \text{Tr}(\boldsymbol{W}_j^T \boldsymbol{Z} \text{softmax}(\beta \; \boldsymbol{Z}^T \boldsymbol{W}_j)) \tag{30}$$

$$= \sum_{s=1}^{S} \sum_{t=1}^{T} \frac{\exp(\beta \boldsymbol{z}_s \boldsymbol{w}_{jt})}{\sum_{s'=1}^{S} \sum_{t'=1}^{T} \exp(\beta \boldsymbol{z}_{s'}^T \boldsymbol{w}_{jt'})} \boldsymbol{z}_s \boldsymbol{w}_{jt} \tag{31}$$

$$= \frac{\sum_{s=1}^{S} \sum_{t=1}^{T} \exp(\beta \boldsymbol{z}_s \boldsymbol{w}_{jt}) \boldsymbol{z}_s \boldsymbol{w}_{jt}}{\sum_{s=1}^{S} \sum_{t=1}^{T} \exp(\beta \boldsymbol{z}_s^T \boldsymbol{w}_{jt})} \tag{32}$$

$$= \frac{\sum_{s=1}^{S} \sum_{t=1}^{T} k(\boldsymbol{z}_s, \boldsymbol{w}_{jt})}{\sum_{s=1}^{S} \sum_{t=1}^{T} k'(\boldsymbol{z}_s, \boldsymbol{w}_{jt})}, \tag{33}$$

where we used the kernels $k, k' : \mathbb{R}^D \times \mathbb{R}^D \to \mathbb{R}$ with feature maps $\varphi : \mathbb{R}^D \to \mathcal{H}$, $\varphi' : \mathbb{R}^D \to \mathcal{H}'$, and $\mathcal{H}, \mathcal{H}'$ denote Hilbert spaces. Given $\boldsymbol{a}, \boldsymbol{b} \in \mathbb{R}^D$ we have

$$k'(\boldsymbol{a}, \boldsymbol{b}) := \exp(\beta \boldsymbol{a}^T \boldsymbol{b}) = \langle \varphi'(\boldsymbol{a}), \varphi'(\boldsymbol{b}) \rangle_{\mathcal{H}'}, \tag{34}$$

$$k(\boldsymbol{a}, \boldsymbol{b}) := k'(\boldsymbol{a}, \boldsymbol{b}) \boldsymbol{a}^T \boldsymbol{b} = \exp(\beta \boldsymbol{a}^T \boldsymbol{b}) \boldsymbol{a}^T \boldsymbol{b} = \langle \varphi(\boldsymbol{a}), \varphi(\boldsymbol{b}) \rangle_{\mathcal{H}}. \tag{35}$$

We define $\bar{\boldsymbol{z}} \in \mathcal{H}$, $\bar{\boldsymbol{w}}_j \in \mathcal{H}$, $\bar{\boldsymbol{z}}' \in \mathcal{H}'$, and $\bar{\boldsymbol{w}}_j' \in \mathcal{H}'$ as follows.

$$\bar{\boldsymbol{z}} := \frac{1}{S} \sum_{s=1}^{S} \varphi(\boldsymbol{z}_s) \quad \bar{\boldsymbol{w}}_j := \frac{1}{T} \sum_{t=1}^{T} \varphi(\boldsymbol{w}_{jt}) \quad \bar{\boldsymbol{z}}' := \frac{1}{S} \sum_{s=1}^{S} \varphi'(\boldsymbol{z}_s) \quad \bar{\boldsymbol{w}}_j' := \frac{1}{T} \sum_{t=1}^{T} \varphi'(\boldsymbol{w}_{jt}) \tag{36}$$

$s_j$ can now be expressed as

$$s_j = \frac{\sum_{s=1}^{S} \sum_{t=1}^{T} k(\boldsymbol{z}_s, \boldsymbol{w}_{jt})}{\sum_{s=1}^{S} \sum_{t=1}^{T} k'(\boldsymbol{z}_s, \boldsymbol{w}_{jt})} \tag{37}$$

$$= \frac{\langle \bar{\boldsymbol{z}}, \bar{\boldsymbol{w}}_j \rangle_{\mathcal{H}}}{\langle \bar{\boldsymbol{z}}', \bar{\boldsymbol{w}}_j' \rangle_{\mathcal{H}'}} \tag{38}$$

We next investigate the term $\mu_j := \text{Tr}(\boldsymbol{W}_j^T \tilde{\boldsymbol{Z}} \text{softmax}(\beta \; \tilde{\boldsymbol{Z}}^T \boldsymbol{W}_j))$. Recall that $\tilde{\boldsymbol{Z}} := (\boldsymbol{Z}_1 \| \cdots \| \boldsymbol{Z}_N)$ where $\boldsymbol{Z}_i := \phi_{\text{enc}}(\boldsymbol{x}_i)$ with $\boldsymbol{Z}_i \in \mathcal{Z}$ is the sequence representation of sequence $\boldsymbol{x}_i$. $\boldsymbol{Z}_i$ is a sequence of stacked tokens $(\boldsymbol{z}_{i1}, \ldots, \boldsymbol{z}_{iS})$, and for simplicity, we assume a uniform sequence length $S$ for all $\boldsymbol{Z}_i$ with $i \in \{1, \ldots, N\}$. We define $\bar{\boldsymbol{z}}_i \in \mathcal{H}$, $\bar{\boldsymbol{z}}_i' \in \mathcal{H}'$, $\boldsymbol{\mu} \in \mathcal{H}$, and $\boldsymbol{\mu}' \in \mathcal{H}'$ as follows:

$$\bar{\boldsymbol{z}}_i := \frac{1}{S} \sum_{s=1}^{S} \varphi(\boldsymbol{z}_{is}) \qquad\qquad \bar{\boldsymbol{z}}_i' := \frac{1}{S} \sum_{s=1}^{S} \varphi'(\boldsymbol{z}_{is}) \tag{39}$$

$$\boldsymbol{\mu} := \frac{1}{N} \sum_{i=1}^{N} \bar{\boldsymbol{z}}_i = \frac{1}{SN} \sum_{i=1}^{N} \sum_{s=1}^{S} \varphi(\boldsymbol{z}_{is}) \tag{40}$$

$$\boldsymbol{\mu}' := \frac{1}{N} \sum_{i=1}^{N} \bar{\boldsymbol{z}}_i = \frac{1}{SN} \sum_{i=1}^{N} \sum_{s=1}^{S} \varphi'(\boldsymbol{z}_{is}) \tag{41}$$

Thus, $\mu_j$ can now be expressed as

$$\mu_j := \mathrm{Tr}(\boldsymbol{W}_j^T \tilde{\boldsymbol{Z}} \mathrm{softmax}(\beta \; \tilde{\boldsymbol{Z}}^T \boldsymbol{W}_j)) \tag{42}$$

$$= \frac{\langle \boldsymbol{\mu}, \bar{\boldsymbol{w}}_j \rangle_{\mathcal{H}}}{\langle \boldsymbol{\mu}', \bar{\boldsymbol{w}}_j' \rangle_{\mathcal{H}'}} \tag{43}$$

$$= \frac{\sum_{i=1}^N \sum_{s=1}^S \sum_{t=1}^T k(\boldsymbol{z}_{is}, \boldsymbol{w}_{jt})}{\sum_{i=1}^N \sum_{s=1}^S \sum_{t=1}^T k'(\boldsymbol{z}_{is}, \boldsymbol{w}_{jt})} \tag{44}$$

We can now express $d^2$ as follows.

$$d^2 = \sum_{j=1}^M \left( \mathrm{Tr}(\boldsymbol{W}_j^T \boldsymbol{Z} \mathrm{softmax}(\beta \; \boldsymbol{Z}^T \boldsymbol{W}_j)) - \mathrm{Tr}(\boldsymbol{W}_j^T \tilde{\boldsymbol{Z}} \mathrm{softmax}(\beta \; \tilde{\boldsymbol{Z}}^T \boldsymbol{W}_j)) \right)^2 \tag{45}$$

$$= \sum_{j=1}^M \left( \frac{\langle \bar{\boldsymbol{z}}, \bar{\boldsymbol{w}}_j \rangle_{\mathcal{H}}}{\langle \bar{\boldsymbol{z}}', \bar{\boldsymbol{w}}_j' \rangle_{\mathcal{H}'}} - \frac{\langle \boldsymbol{\mu}, \bar{\boldsymbol{w}}_j \rangle_{\mathcal{H}}}{\langle \boldsymbol{\mu}', \bar{\boldsymbol{w}}_j' \rangle_{\mathcal{H}'}} \right)^2 \tag{46}$$

Therefore, $d^2$ is a modified version of the "idealized" distance $d_{\mathcal{H}}^2$:

$$d_{\mathcal{H}}^2 = \sum_{j=1}^M \left( \langle \bar{\boldsymbol{z}}, \bar{\boldsymbol{w}}_j \rangle_{\mathcal{H}} - \langle \boldsymbol{\mu}, \bar{\boldsymbol{w}}_j \rangle_{\mathcal{H}} \right)^2 \tag{47}$$

$$= (\bar{\boldsymbol{z}} - \boldsymbol{\mu})^T \left( \sum_{j=1}^M \bar{\boldsymbol{w}}_j \bar{\boldsymbol{w}}_j^T \right) (\bar{\boldsymbol{z}} - \boldsymbol{\mu}). \tag{48}$$

In early experiments, we evaluated $d_{\mathcal{H}}^2$ using a range of kernel functions. It performed substantially worse than the $d^2$ with attention pooling. This might be due to gradients being either too large (e.g., with the exponential kernel) or too small (e.g., with the RBF kernel) for gradient-based learning.

## B.4 AP-OOD Reduces to Mahalanobis Distance with Mean Pooling for $\beta = 0$

In this section, we show that as $\beta = 0$ and $M = D$, $d^2(\boldsymbol{Z}, \tilde{\boldsymbol{Z}})$ reduces to the Mahalanobis distance with mean pooling as used by Ren et al. (2023). To arrive at the result, we assume uniform sequence lengths.

$$\mathrm{softmax}(0 \cdot \boldsymbol{Z}^T \boldsymbol{w})_s = \frac{\exp(0 \cdot \boldsymbol{z}_s^T \boldsymbol{w})}{\sum_{s'=1}^S \exp(0 \cdot \boldsymbol{z}_{s'}^T \boldsymbol{w})} = \frac{1}{S}, \tag{49}$$

$$\bar{\boldsymbol{z}} = \mathrm{AttPool}_0(\boldsymbol{Z}, \boldsymbol{w}) = \boldsymbol{Z} \mathrm{softmax}(0 \cdot \boldsymbol{Z}^T \boldsymbol{w}) = \frac{1}{S} \sum_{s=1}^S \boldsymbol{z}_s, \tag{50}$$

$$\boldsymbol{\mu} = \mathrm{AttPool}_0(\tilde{\boldsymbol{Z}}, \boldsymbol{w}) = \tilde{\boldsymbol{Z}} \mathrm{softmax}(0 \cdot \tilde{\boldsymbol{Z}}^T \boldsymbol{w}) = \frac{1}{SN} \sum_{i=1}^N \sum_{s=1}^S \boldsymbol{z}_{is} = \frac{1}{N} \sum_{i=1}^N \bar{\boldsymbol{z}}_i, \tag{51}$$

where we use the concatenated sequence $\tilde{\boldsymbol{Z}} = (\boldsymbol{Z}_1 \| \cdots \| \boldsymbol{Z}_N)$, and the sequence representations $\boldsymbol{Z}_i = \phi(\boldsymbol{x}_i) = (\boldsymbol{z}_{i1}, \ldots, \boldsymbol{z}_{iS}) \in \mathbb{R}^{D \times S}$. The squared distance of AP-OOD reduces to

$$d^2(\boldsymbol{Z}, \tilde{\boldsymbol{Z}}) = \sum_{j=1}^M \left( \boldsymbol{w}_j^T \boldsymbol{Z} \mathrm{softmax}(\beta \; \boldsymbol{Z}^T \boldsymbol{w}_j) - \boldsymbol{w}_j^T \tilde{\boldsymbol{Z}} \mathrm{softmax}(\beta \; \tilde{\boldsymbol{Z}}^T \boldsymbol{w}_j) \right)^2 \tag{52}$$

$$= \sum_{j=1}^D (\boldsymbol{w}_j^T \bar{\boldsymbol{z}} - \boldsymbol{w}_j^T \boldsymbol{\mu})^2 = d_{\mathrm{Maha}}^2(\bar{\boldsymbol{z}}, \boldsymbol{\mu}). \tag{53}$$

## C    ADDITIONAL ALGORITHMIC DETAILS

### C.1    MINI-BATCH ATTENTION POOLING

In this section, we describe the process of performing attention pooling over a long sequence representation $\tilde{\boldsymbol{Z}} \in \mathcal{Z}$ that is too large to fit into memory by dividing $\tilde{\boldsymbol{Z}}$ into smaller mini-batches of size $B \in \mathbb{N}$. For this, we need the log-sum-exponential (lse) function. We follow the notation from Ramsauer et al. (2021):

$$\mathrm{lse}(\beta, \boldsymbol{a}) \;=\; \beta^{-1} \log \left( \sum_{s=1}^{S} \exp(\beta a_s) \right) \tag{54}$$

The following algorithm computes $\boldsymbol{\mu} = \tilde{\boldsymbol{Z}}\mathrm{softmax}(\beta \; \tilde{\boldsymbol{Z}}^T \boldsymbol{w})$ for $\beta > 0$, and $\sigma$ denotes the logistic sigmoid function:

---

**Algorithm 2** Attention pooling over a long sequence

---

**Require:** $\tilde{\boldsymbol{Z}} = (\tilde{\boldsymbol{z}}_1, \ldots, \tilde{\boldsymbol{z}}_S) \in \mathbb{R}^{D \times S}$, inverse temperature $\beta$, weight vector $\boldsymbol{w}$, batch size $B$
1: $E \leftarrow -\infty$
2: $\boldsymbol{\mu} \leftarrow \boldsymbol{0}$
3: **for** $s \leftarrow 1$ to $S$ **step** $B$ **do**
4:     Load mini-batch $\boldsymbol{B} \leftarrow (\tilde{\boldsymbol{z}}_s, \ldots, \tilde{\boldsymbol{z}}_{s+B})$
5:     $E_B \leftarrow \mathrm{lse}(\beta, \boldsymbol{B}^T \boldsymbol{w})$
6:     $\boldsymbol{p} \leftarrow \exp(\beta(\boldsymbol{B}^T \boldsymbol{w} - E_B))$
7:     $\boldsymbol{\mu}_B \leftarrow \boldsymbol{B}\boldsymbol{p}$
8:     $p_B \leftarrow \sigma(\beta(E_B - E))$
9:     $\boldsymbol{\mu} \leftarrow p_B \boldsymbol{\mu}_B + (1 - p_B)\boldsymbol{\mu}$
10:     $E \leftarrow \beta^{-1} \log\left( \exp(\beta E_B) + \exp(\beta E) \right)$
    **return** $\boldsymbol{\mu}$

---

### C.2    AP-OOD IN PYTORCH/EINOPS-LIKE PSEUDOCODE

We detail the loss computation for AP-OOD with multiple queries per head (Equation (14)) using PyTorch/Einops-style pseudocode. Assuming a uniform sequence length $S$, Algorithm 3 demonstrates the computation via attention pooling over the sequences $\mathbf{Z}$. Alternatively, Algorithm 4 presents a mathematically equivalent formulation that applies attention pooling over the similarities $\mathbf{W}^T \mathbf{Z}$.

### C.3    ON THE DIFFERENCE BETWEEN HEADS AND QUERIES

We find that heads are learnt largely independently from one another while queries are not, which we experimentally verify as follows: We train AP-OOD using the SGD optimizer on the summarization task using (i) 1 head and (ii) 2 heads, where the initialization of one of the heads in (ii) is identical to the initialization of the head of (i). We find that after training for 500 steps, the weight vectors associated with the heads with shared initialization between (i) and (ii) remain identical. In contrast, when repeating this experiment by varying the number of queries, the weight vectors associated with the queries differ after training. Intuitively, adding additional heads will help the model discover more local minima in the parameter space (similar to Lakshminarayanan et al., 2017), while adding queries increases the capacity of each given head. The following observation supports this intuition: When testing different hyperparameter combinations for AP-OOD, we found that a large number of queries combined with a small number of heads leads to overfitting when training the model on small ID data sets (e.g., 10,000 sequences): In this case, the average distance of the ID training sequences is substantially smaller than the average distance of the ID validation sequences.

**Algorithm 3** AP-OOD loss in PyTorch/Einops-like style using attention pooling over $\boldsymbol{Z}$

```python
def attention_pooling(Zs, Ws):
    # Zs[N S D] - mini-batch of N sequences with length S
    # Ws[M T D] - weights of model with M heads and T queries

    # pairwise similarities between tokens and weights
    similarities = einsum(Zs, Ws, 'N S D, M T D -> N M S T')

    # softmax over query- and sequence dimensions
    probs = similarities.softmax(dim=(2, 3)) #[N M S T]

    # pooling to form new sequences
    Z_bars = einsum(Zs, probs, 'N S D, N M S T ->N M T D')

    return Z_bars

def loss(Zs, Ws):
    # Zs[N S D] - mini-batch of N sequences with length S
    # Ws[M T D] - weights of model with M heads and T queries

    # attention pooling over individual sequence
    Z_bars = attention_pooling(Zs, Ws) #[N M T D]

    # attention pooling over all sequences
    Z_tilde = Zs.flatten(0, 1).unsqueeze(0) #[1, N*S, D]
    mus = attention_pooling(Z_tilde, Ws) #[1 M T D]

    # squared distance per head
    heads_Z = einsum(Z_bars, Ws, 'N M T D, M T D -> N M')
    heads_mu = einsum(mus, Ws, '1 M T D, M T D -> 1 M')
    ds_squared = (heads_Z - heads_mu)**2 #[N M]

    # regularized and loss
    regularizer = torch.log((Ws * Ws).sum(dim=(1, 2))) #[M]
    losses = (ds_squared - regularizer).sum(dim=1) #[N]
    loss = torch.mean(losses)

    return loss
```

## D EXPERIMENTS

### D.1 ADDITIONAL DETAILS FOR THE TOY EXPERIMENT

In the toy experiment in Figure 2, we modify the attention pooling process to use the negative squared Euclidean distance instead of the dot product similarity because the Euclidean distance is known to work better in low-dimensional spaces. Formally, the modified attention pooling process is:

$$\text{AttPool}_\beta(\boldsymbol{Z}, \boldsymbol{w}) := \sum_{s=1}^{S} \boldsymbol{z}_s \frac{\exp(-\frac{\beta}{2}||\boldsymbol{z}_s - \boldsymbol{w}||_2^2)}{\sum_{s'=1}^{S} \exp(-\frac{\beta}{2}||\boldsymbol{z}_{s'} - \boldsymbol{w}||_2^2)}. \tag{55}$$

### D.2 HYPERPARAMETER SELECTION

To find the values for $\beta$, $M$, and $T$ in the unsupervised setting, we perform a grid search using the values $\beta \in \{\frac{1}{\sqrt{D}}, 0.25, 0.5, 1, 2\}$ and $T \in \{1, 4, 16, 64\}$. We select $M$ such that the total number of parameters of AP-OOD equals the number of entries in $\boldsymbol{\Sigma}$ of the Mahalanobis method, i.e., such that $MT = D$. We select the hyperparameter configuration by evaluating each resulting model on OOD detection using a validation split of the AUX data set (in the unsupervised setting, we use the AUX data set only for model selection, not for training the

**Algorithm 4** AP-OOD loss in PyTorch/Einops-like style using attention pooling over $\boldsymbol{W}^T\boldsymbol{Z}$

```python
def attention_pooling(Zs, Ws):
    # Zs[N S D] - mini-batch of N sequences with length S
    # Ws[M T D] - weights of model with M heads and T queries

    # pairwise similarities between tokens and weights
    similarities = einsum(Zs, Ws, 'N S D, M T D -> N M S T')

    # softmax over query- and sequence dimensions
    probs = similarities.softmax(dim=(2, 3)) #[N M S T]

    # pooling over similarities
    pooled = einsum(similarities, probs 'N M S T, N M S T -> N M')

    return pooled

def loss(Zs, Ws):
    # Zs[N S D] - mini-batch of N sequences with length S
    # Ws[M T D] - weights of model with M heads and T queries

    # attention pooling over individual sequence
    heads_Z = attention_pooling(Zs, Ws) #[N M]

    # attention pooling over all sequences
    Z_tilde = Zs.flatten(0, 1).unsqueeze(0) #[1, N*S, D]
    heads_mus = attention_pooling(Z_tilde, Ws) #[1 M]

    # squared distance per head
    ds_squared = (heads_Z - heads_mu)**2 #[N M]

    # regularizer and loss
    regularizer = torch.log((Ws * Ws).sum(dim=(1, 2))) #[M]
    losses = (ds_squared - regularizer).sum(dim=1) #[N]
    loss = torch.mean(losses)

    return loss
```

model), and we select the model with the highest AUROC. In the supervised setting, we follow the same procedure, and we additionally select $\lambda \in \{0.1, 1, 10\}$.

### D.3 Supervised Experiments on Text Summarization

In the fully supervised setting, we train all methods on the embeddings of 100,000 ID examples and 10,000 AUX examples obtained from PEGASUS$_{\text{LARGE}}$ trained on text summarization using the XSUM data set. Table 4 shows that AP-OOD substantially improves fully supervised OOD detection results, improving the previously best mean FPR95 of 0.97% (binary logits) to 0.11% in the input OOD setting. Figure 5 shows the results for the semi-supervised setting when scaling the number of AUX examples on all OOD data sets for text summarization. We evaluate relative Mahalanobis only for $N' \geq 1024$, because $\boldsymbol{\Sigma}$ is not invertible when using fewer AUX examples. In contrast to Figure 3, Figure 5 also shows the results for Reddit TIFU and Samsum. On these two data sets, all evaluated methods except relative Mahalanobis achieve near-perfect OOD detection results for $N' \geq 8$.

### D.4 Visualizing AP-OOD's Attention Maps on the Summarization Task

We analyze how the attention pooling process of AP-OOD allocates weight to individual tokens. We randomly select one sample from each of the four OOD data sets in the summarization benchmark. We then investigate the attention weights of a trained AP-OOD model over the generated output sequence. For each sample, we select the two heads with the

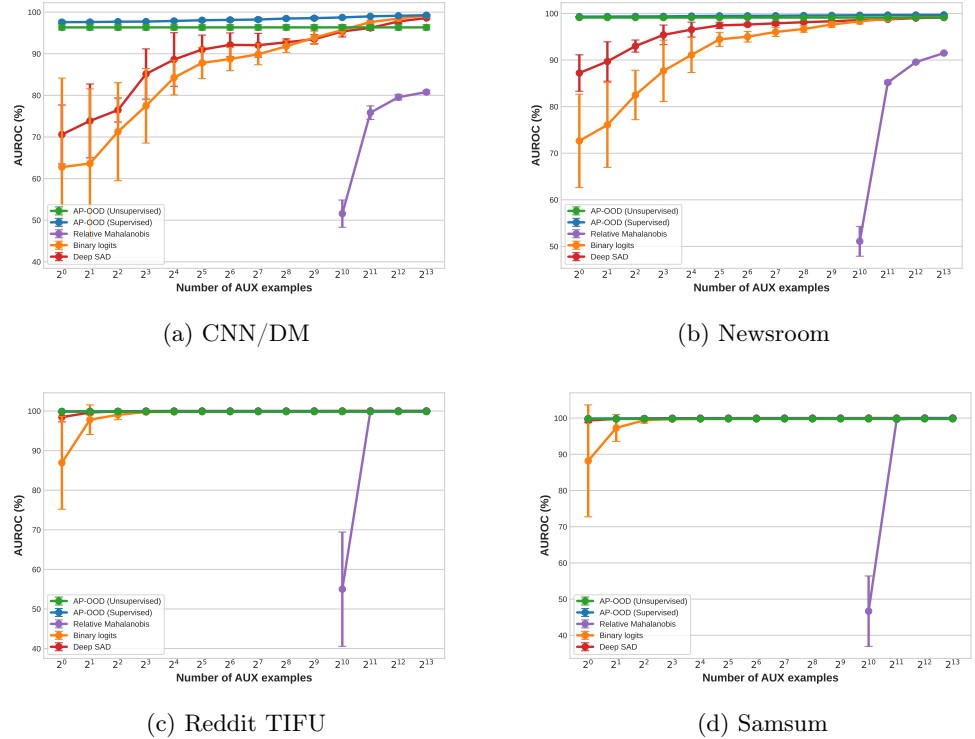

(a) CNN/DM

(b) Newsroom

(c) Reddit TIFU

(d) Samsum

Figure 5: OOD detection performance on text summarization for all OOD data sets. We vary the number of AUX examples and compare results from AP-OOD, binary logits (Ren et al., 2023), relative Mahalanobis (Ren et al., 2023), and Deep SAD (Ruff et al., 2019).

Table 4: Supervised OOD detection performance on text summarization. We compare results from AP-OOD, binary logits (Ren et al., 2023), relative Mahalanobis (Ren et al., 2023), and Deep SAD (Ruff et al., 2019) on PEGASUS$_{\text{LARGE}}$ trained on XSUM as the ID data set. ↓ indicates "lower is better" and ↑ "higher is better". All values in %. We estimate standard deviations across five independent data set splits and training runs.

| | | CNN/DM | Newsroom | Reddit | Samsum | Mean |
|---|---|---|---|---|---|---|
| | | Input OOD | | | | |
| Binary logits | AUROC ↑ | $99.49^{\pm0.07}$ | $99.47^{\pm0.06}$ | $\mathbf{100.00^{\pm0.00}}$ | $99.98^{\pm0.01}$ | $99.74$ |
| | FPR95 ↓ | $1.81^{\pm0.29}$ | $2.04^{\pm0.17}$ | $0.00^{\pm0.00}$ | $0.03^{\pm0.02}$ | $0.97$ |
| Relative Mahalanobis | AUROC ↑ | $81.43^{\pm0.72}$ | $91.71^{\pm0.17}$ | $99.95^{\pm0.01}$ | $\mathbf{99.99^{\pm0.00}}$ | $93.27$ |
| | FPR95 ↓ | $62.60^{\pm0.80}$ | $28.41^{\pm0.34}$ | $0.01^{\pm0.01}$ | $0.01^{\pm0.01}$ | $22.75$ |
| Deep SAD | AUROC ↑ | $99.34^{\pm0.04}$ | $99.44^{\pm0.02}$ | $\mathbf{100.00^{\pm0.00}}$ | $\mathbf{100.00^{\pm0.00}}$ | $99.70$ |
| | FPR95 ↓ | $1.18^{\pm0.13}$ | $1.49^{\pm0.13}$ | $0.00^{\pm0.00}$ | $0.00^{\pm0.00}$ | $0.67$ |
| AP-OOD (Ours) | AUROC ↑ | $\mathbf{99.89^{\pm0.02}}$ | $\mathbf{99.89^{\pm0.02}}$ | $\mathbf{100.00^{\pm0.00}}$ | $\mathbf{99.99^{\pm0.01}}$ | $\mathbf{99.94}$ |
| | FPR95 ↓ | $\mathbf{0.05^{\pm0.01}}$ | $0.39^{\pm0.05}$ | $\mathbf{0.00^{\pm0.00}}$ | $\mathbf{0.00^{\pm0.00}}$ | $\mathbf{0.11}$ |
| | | Output OOD | | | | |
| Binary logits | AUROC ↑ | $98.47^{\pm0.34}$ | $99.45^{\pm0.08}$ | $99.99^{\pm0.00}$ | $99.95^{\pm0.02}$ | $99.47$ |
| | FPR95 ↓ | $5.77^{\pm1.27}$ | $1.87^{\pm0.29}$ | $\mathbf{0.00^{\pm0.00}}$ | $0.04^{\pm0.02}$ | $1.92$ |
| Relative Mahalanobis | AUROC ↑ | $93.42^{\pm0.24}$ | $97.38^{\pm0.08}$ | $99.82^{\pm0.01}$ | $99.52^{\pm0.03}$ | $97.54$ |
| | FPR95 ↓ | $25.22^{\pm0.75}$ | $8.68^{\pm0.25}$ | $0.03^{\pm0.00}$ | $0.95^{\pm0.15}$ | $8.72$ |
| Deep SAD | AUROC ↑ | $97.53^{\pm0.22}$ | $99.40^{\pm0.08}$ | $99.99^{\pm0.00}$ | $99.95^{\pm0.00}$ | $99.22$ |
| | FPR95 ↓ | $8.22^{\pm0.57}$ | $1.90^{\pm0.22}$ | $0.01^{\pm0.01}$ | $0.08^{\pm0.03}$ | $2.55$ |
| AP-OOD (Ours) | AUROC ↑ | $\mathbf{99.46^{\pm0.07}}$ | $\mathbf{99.73^{\pm0.03}}$ | $\mathbf{100.00^{\pm0.00}}$ | $\mathbf{99.99^{\pm0.00}}$ | $\mathbf{99.80}$ |
| | FPR95 ↓ | $1.96^{\pm0.31}$ | $0.97^{\pm0.09}$ | $\mathbf{0.00^{\pm0.00}}$ | $\mathbf{0.00^{\pm0.00}}$ | $\mathbf{0.73}$ |

largest deviations in the positive and in the negative directions before applying the square in the score function of AP-OOD. Figure 6 visualizes the token-wise attention weights of the selected heads. When manually examining the generated output sequences, we find it hard to attribute the "OODness" of individual sequences to a single token or to a small set

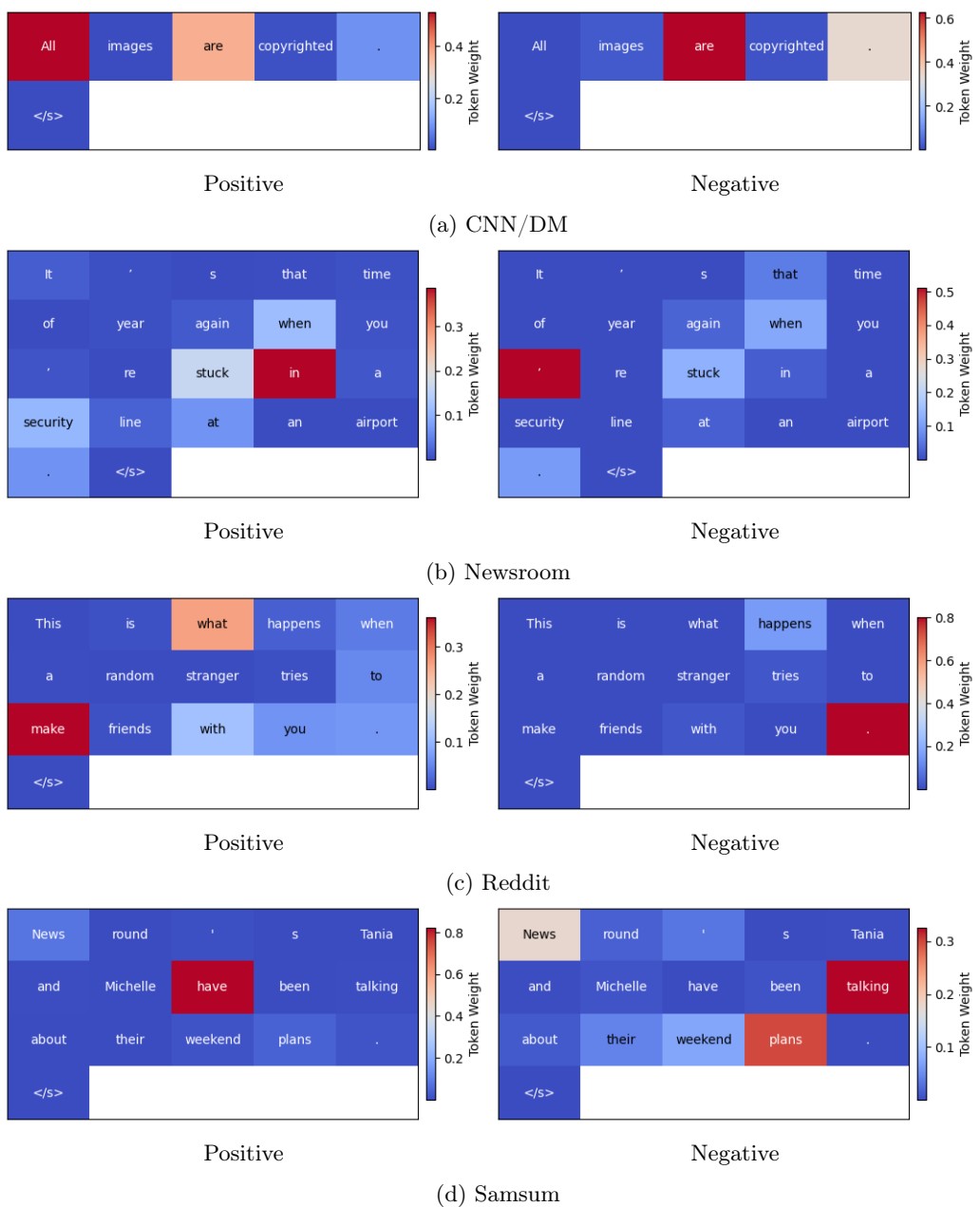

(a) CNN/DM

(b) Newsroom

(c) Reddit

(d) Samsum

Figure 6: AP-OOD's attention weights on randomly selected output sequences from OOD data sets on text summarization. For each sequence, we visualize the heads $j$ with the highest deviation in the positive and negative direction of the $d_j(\boldsymbol{Z})$ before applying the square.

of tokens. Therefore, it is difficult to interpret the attention weights for the individual heads. However, the results indicate that the different heads exhibit distinct attention patterns.

## D.5 Supervised Experiments on Translation

In the fully supervised setting, we train all methods on the embeddings of 100,000 ID embeddings and 100,000 AUX embeddings obtained from a Transformer (base) trained on WMT15 En–Fr translation. Table 5 shows that AP-OOD improves supervised OOD detection results w.r.t. the mean AUROC and mean FPR95 metrics.

Table 5: Supervised OOD detection performance on English-to-French translation. We compare results from AP-OOD, binary logits, relative mahalanobis (Ren et al., 2023), and Deep SAD (Ruff et al., 2019) on a Transformer (base) trained on WMT15 En–Fr as the ID data set. ↓ indicates "lower is better" and ↑ "higher is better". All values in %. We estimate standard deviations across five independent data set splits and training runs.

| | | IT | Koran | Law | Medical | Subtitles | ndd2015 | ndt2015 | nt2014 | Mean |
|---|---|---|---|---|---|---|---|---|---|---|
| | | | | | Input OOD | | | | | |
| Binary logits | AUROC ↑ | $95.42^{\pm0.71}$ | $96.41^{\pm0.19}$ | $51.23^{\pm8.43}$ | $73.81^{\pm1.89}$ | $91.56^{\pm0.73}$ | $90.59^{\pm0.58}$ | $90.50^{\pm0.39}$ | $88.39^{\pm0.47}$ | 84.74 |
| | FPR95 ↓ | $20.51^{\pm2.56}$ | $22.61^{\pm2.14}$ | $95.96^{\pm0.80}$ | $80.14^{\pm1.60}$ | $41.74^{\pm2.03}$ | $\underline{57.18}^{\pm2.41}$ | $\underline{55.63}^{\pm1.90}$ | $\underline{68.56}^{\pm1.19}$ | 55.29 |
| Relative Mahalanobis | AUROC ↑ | $88.21^{\pm0.97}$ | $96.17^{\pm0.31}$ | $30.11^{\pm2.34}$ | $67.20^{\pm0.73}$ | $90.98^{\pm0.89}$ | $88.62^{\pm0.42}$ | $87.63^{\pm0.48}$ | $84.76^{\pm0.48}$ | 79.21 |
| | FPR95 ↓ | $29.06^{\pm2.24}$ | $22.13^{\pm1.82}$ | $96.99^{\pm0.34}$ | $\mathbf{79.45^{\pm1.60}}$ | $\mathbf{26.71^{\pm1.65}}$ | $67.87^{\pm0.65}$ | $67.83^{\pm0.87}$ | $82.76^{\pm0.58}$ | 59.10 |
| Deep SAD | AUROC ↑ | $\underline{97.19}^{\pm0.22}$ | $96.74^{\pm0.12}$ | $\underline{52.11}^{\pm2.30}$ | $\underline{80.55}^{\pm0.68}$ | $93.85^{\pm0.45}$ | $\underline{91.11}^{\pm0.34}$ | $\underline{90.99}^{\pm0.32}$ | $88.27^{\pm0.34}$ | $\underline{86.35}$ |
| | FPR95 ↓ | $\underline{17.37}^{\pm1.41}$ | $\underline{20.06}^{\pm1.23}$ | $\underline{95.95}^{\pm0.35}$ | $81.62^{\pm1.37}$ | $39.00^{\pm2.65}$ | $57.40^{\pm1.91}$ | $57.33^{\pm1.68}$ | $70.96^{\pm0.88}$ | $\underline{54.96}$ |
| AP-OOD (Ours) | AUROC ↑ | $\mathbf{97.41^{\pm0.19}}$ | $\mathbf{97.98^{\pm0.19}}$ | $\mathbf{60.30^{\pm4.59}}$ | $\mathbf{81.28^{\pm1.04}}$ | $\mathbf{95.41^{\pm0.49}}$ | $\mathbf{92.32^{\pm0.34}}$ | $\mathbf{91.89^{\pm0.40}}$ | $\mathbf{89.91^{\pm0.30}}$ | $\mathbf{88.31}$ |
| | FPR95 ↓ | $\mathbf{16.89^{\pm1.33}}$ | $\mathbf{11.63^{\pm0.73}}$ | $\mathbf{95.91^{\pm0.61}}$ | $\underline{79.79}^{\pm1.55}$ | $\underline{29.25}^{\pm4.15}$ | $\mathbf{49.03^{\pm2.26}}$ | $\mathbf{49.53^{\pm1.46}}$ | $\mathbf{63.15^{\pm2.04}}$ | $\mathbf{49.40}$ |
| | | | | | Output OOD | | | | | |
| Binary logits | AUROC ↑ | $94.40^{\pm0.36}$ | $96.73^{\pm0.23}$ | $54.28^{\pm4.88}$ | $72.91^{\pm0.97}$ | $92.45^{\pm0.68}$ | $89.98^{\pm0.40}$ | $89.91^{\pm0.41}$ | $87.39^{\pm0.43}$ | 84.76 |
| | FPR95 ↓ | $28.68^{\pm1.40}$ | $\underline{19.42}^{\pm2.09}$ | $96.42^{\pm0.70}$ | $83.66^{\pm2.03}$ | $37.40^{\pm2.33}$ | $\underline{56.02}^{\pm1.30}$ | $56.99^{\pm2.05}$ | $70.80^{\pm1.24}$ | 56.17 |
| Relative Mahalanobis | AUROC ↑ | $87.68^{\pm1.03}$ | $95.86^{\pm0.21}$ | $30.41^{\pm1.60}$ | $66.17^{\pm0.95}$ | $92.42^{\pm0.61}$ | $88.35^{\pm0.42}$ | $87.70^{\pm0.47}$ | $84.17^{\pm0.42}$ | 79.09 |
| | FPR95 ↓ | $37.33^{\pm2.12}$ | $24.31^{\pm1.33}$ | $96.72^{\pm0.16}$ | $82.28^{\pm1.45}$ | $\underline{30.83}^{\pm1.94}$ | $69.29^{\pm1.36}$ | $68.53^{\pm1.46}$ | $84.15^{\pm0.83}$ | 61.68 |
| Deep SAD | AUROC ↑ | $91.49^{\pm0.36}$ | $76.54^{\pm1.43}$ | $\mathbf{63.38^{\pm1.57}}$ | $\underline{77.14}^{\pm0.27}$ | $84.72^{\pm0.44}$ | $64.46^{\pm0.78}$ | $65.57^{\pm0.70}$ | $51.65^{\pm0.52}$ | 71.87 |
| | FPR95 ↓ | $51.80^{\pm1.30}$ | $91.73^{\pm1.03}$ | $91.53^{\pm0.76}$ | $72.54^{\pm0.87}$ | $80.49^{\pm0.94}$ | $93.97^{\pm0.43}$ | $93.07^{\pm0.22}$ | $98.35^{\pm0.13}$ | 84.18 |
| AP-OOD (Ours) | AUROC ↑ | $\mathbf{97.14^{\pm0.19}}$ | $\mathbf{97.71^{\pm0.17}}$ | $\underline{56.60}^{\pm2.72}$ | $\mathbf{81.44^{\pm0.85}}$ | $\mathbf{95.27^{\pm0.50}}$ | $\mathbf{91.49^{\pm0.47}}$ | $\mathbf{91.28^{\pm0.47}}$ | $\mathbf{89.06^{\pm0.30}}$ | $\mathbf{87.50}$ |
| | FPR95 ↓ | $\mathbf{19.52^{\pm1.08}}$ | $\mathbf{13.73^{\pm1.85}}$ | $97.18^{\pm0.38}$ | $\underline{80.53}^{\pm0.42}$ | $\mathbf{30.02^{\pm4.17}}$ | $\mathbf{51.87^{\pm3.13}}$ | $\mathbf{50.87^{\pm2.52}}$ | $\mathbf{66.31^{\pm1.75}}$ | $\mathbf{51.25}$ |

Table 6: Unsupervised OOD detection performance on large-scale language modeling. We compare results from AP-OOD, Mahalanobis (Lee et al., 2018; Ren et al., 2023), KNN (Sun et al., 2022), DeepSVDD (Ruff et al., 2018), and model perplexity (Ren et al., 2023) on Pythia-160M trained on the Pile as the ID data set. ↓ indicates "lower is better" and ↑ "higher is better". All values in %. Standard deviations are estimated across five independent training runs.

| | | 4Chan | Reports | Covid | Clinical | Twitter | Mean |
|---|---|---|---|---|---|---|---|
| | | | | Input OOD | | | |
| Perplexity | AUROC ↑ | $\underline{65.05}$ | 48.32 | $\underline{89.51}$ | $\underline{85.86}$ | $\mathbf{99.22}$ | $\underline{77.59}$ |
| | FPR95 ↓ | $\mathbf{72.66}$ | $\mathbf{86.91}$ | 68.60 | 65.22 | $\underline{2.85}$ | $\underline{59.25}$ |
| Mahalanobis | AUROC ↑ | 35.27 | 54.72 | 75.50 | 75.67 | 97.86 | 67.81 |
| | FPR95 ↓ | 92.93 | $\underline{87.77}$ | 98.07 | 90.79 | 10.91 | 76.09 |
| KNN | AUROC ↑ | 39.31 | 59.41 | 70.62 | 75.56 | 81.50 | 65.28 |
| | FPR95 ↓ | 98.85 | 93.26 | 99.03 | 95.85 | 61.12 | 89.62 |
| Deep SVDD | AUROC ↑ | 55.59 | $\underline{64.06}$ | 72.54 | 73.44 | 81.08 | 69.34 |
| | FPR95 ↓ | $\underline{88.15}$ | 88.94 | 99.52 | 96.35 | 76.72 | 89.93 |
| AP-OOD (Ours) | AUROC ↑ | $\mathbf{87.97}$ | $\mathbf{68.09}$ | $\mathbf{91.79}$ | $\mathbf{86.44}$ | $\underline{99.08}$ | $\mathbf{86.67}$ |
| | FPR95 ↓ | 88.34 | 91.47 | $\mathbf{40.34}$ | $\mathbf{57.38}$ | $\mathbf{1.52}$ | $\mathbf{55.81}$ |

## D.6 EXPERIMENTS ON DECODER-ONLY LANGUAGE MODELING

To verify the effectiveness of AP-OOD on the decoder-only language modeling paradigm used by LLMs, we conduct experiments on Pythia-160M (Biderman et al., 2023), a decoder-only language model trained on the Pile (Gao et al., 2020). We evaluate the discriminative power of AP-OOD trained in an unsupervised fashion on the 4Chan and Twitter subsets of Paloma (Magnusson et al., 2024), the EDGAR annual reports corpus (annual reports of public companies between 1993–2020; Loukas et al., 2021), Long-COVID related articles (Langnickel et al., 2022), and the MIMIC-III clinical corpus (Goldberger et al., 2000). In the decoder-only setting, we directly use the encoded representations of the input sequences and do not generate output sequences. Table 6 shows that AP-OOD improves unsupervised OOD detection w.r.t. the mean AUROC and mean FPR95 metrics.

## D.7 OOD SCORE COMPARISON

We experimentally compare the min-based OOD score $s_{\min}(\boldsymbol{Z})$ and its upper bound $s(\boldsymbol{Z})$. For training, we use the loss from Equation (10) in both settings. The results in Table 7

Table 7: Unsupervised OOD detection performance on text summarization. We compare results from AP-OOD when using $s(\boldsymbol{Z})$ and $s_{\min}(\boldsymbol{Z})$, on PEGASUS$_{\text{LARGE}}$ trained on XSUM as the ID data set. ↓ indicates "lower is better" and ↑ "higher is better". All values in %. We estimate standard deviations across five independent dataset splits and training runs.

| | | CNN/DM | Newsroom | Reddit | Samsum | Mean |
|---|---|---|---|---|---|---|
| | | | Input OOD | | | |
| $s(\boldsymbol{Z})$ | AUROC ↑ | $\mathbf{96.13}^{\pm \mathbf{0.44}}$ | $\mathbf{99.10}^{\pm \mathbf{0.08}}$ | $\mathbf{99.91}^{\pm \mathbf{0.03}}$ | $\mathbf{99.80}^{\pm \mathbf{0.04}}$ | **98.74** |
| | FPR95 ↓ | $19.51^{\pm 2.24}$ | $\mathbf{4.11}^{\pm \mathbf{0.28}}$ | $\mathbf{0.00}^{\pm \mathbf{0.01}}$ | $\mathbf{0.04}^{\pm \mathbf{0.03}}$ | **5.91** |
| $s_{\min}(\boldsymbol{Z})$ | AUROC ↑ | $96.08^{\pm 0.37}$ | $97.48^{\pm 0.28}$ | $99.71^{\pm 0.20}$ | $97.67^{\pm 0.35}$ | 97.74 |
| | FPR95 ↓ | $\mathbf{18.78}^{\pm \mathbf{2.73}}$ | $11.16^{\pm 1.21}$ | $0.01^{\pm 0.01}$ | $12.04^{\pm 3.04}$ | 10.50 |
| | | | Output OOD | | | |
| $s(\boldsymbol{Z})$ | AUROC ↑ | $93.37^{\pm 0.54}$ | $\mathbf{92.62}^{\pm \mathbf{0.67}}$ | $\mathbf{98.04}^{\pm \mathbf{0.28}}$ | $\mathbf{98.30}^{\pm \mathbf{0.11}}$ | **95.59** |
| | FPR95 ↓ | $23.12^{\pm 1.97}$ | $\mathbf{29.91}^{\pm \mathbf{2.93}}$ | $\mathbf{6.34}^{\pm \mathbf{1.56}}$ | $\mathbf{6.83}^{\pm \mathbf{0.64}}$ | **16.55** |
| $s_{\min}(\boldsymbol{Z})$ | AUROC ↑ | $\mathbf{93.82}^{\pm \mathbf{1.56}}$ | $88.30^{\pm 3.45}$ | $95.94^{\pm 2.25}$ | $90.13^{\pm 4.31}$ | 92.05 |
| | FPR95 ↓ | $26.60^{\pm 5.53}$ | $38.26^{\pm 3.73}$ | $18.49^{\pm 9.01}$ | $36.71^{\pm 12.40}$ | 30.02 |

Table 8: Unsupervised OOD detection performance on text summarization. We compare results from AP-OOD trained on XSUM as the ID data set when varying $\beta$. ↓ indicates "lower is better" and ↑ "higher is better". All values in %. We estimate standard deviations across five independent dataset splits and training runs.

| | | CNN/DM | Newsroom | Reddit | Samsum | Mean |
|---|---|---|---|---|---|---|
| | | | Input OOD | | | |
| $\beta = 0$ | AUROC ↑ | $66.83^{\pm 0.44}$ | $81.42^{\pm 0.27}$ | $94.81^{\pm 0.32}$ | $93.38^{\pm 0.20}$ | 84.11 |
| | FPR95 ↓ | $97.17^{\pm 0.10}$ | $76.31^{\pm 0.35}$ | $41.12^{\pm 3.42}$ | $19.96^{\pm 0.84}$ | 58.64 |
| $\beta = 0.25$ | AUROC ↑ | $\mathbf{97.76}^{\pm \mathbf{0.11}}$ | $98.75^{\pm 0.07}$ | $\underline{99.87}^{\pm 0.06}$ | $99.46^{\pm 0.09}$ | **98.96** |
| | FPR95 ↓ | $\mathbf{11.07}^{\pm \mathbf{0.74}}$ | $\underline{4.75}^{\pm 0.41}$ | $\mathbf{0.00}^{\pm \mathbf{0.00}}$ | $\underline{0.02}^{\pm 0.02}$ | **3.96** |
| $\beta = 0.5$ | AUROC ↑ | $96.13^{\pm 0.44}$ | $99.10^{\pm 0.08}$ | $\mathbf{99.91}^{\pm \mathbf{0.03}}$ | $99.80^{\pm 0.04}$ | $\underline{98.74}$ |
| | FPR95 ↓ | $\underline{19.51}^{\pm 2.24}$ | $\mathbf{4.11}^{\pm \mathbf{0.28}}$ | $0.00^{\pm 0.01}$ | $0.04^{\pm 0.03}$ | $\underline{5.91}$ |
| $\beta = 1$ | AUROC ↑ | $91.36^{\pm 0.41}$ | $\underline{98.77}^{\pm 0.05}$ | $99.75^{\pm 0.02}$ | $\underline{99.83}^{\pm 0.01}$ | 97.43 |
| | FPR95 ↓ | $38.78^{\pm 4.50}$ | $4.94^{\pm 0.23}$ | $0.02^{\pm 0.02}$ | $\mathbf{0.00}^{\pm \mathbf{0.00}}$ | 10.94 |
| $\beta = 2$ | AUROC ↑ | $84.29^{\pm 0.91}$ | $97.58^{\pm 0.09}$ | $99.52^{\pm 0.05}$ | $99.76^{\pm 0.01}$ | 95.28 |
| | FPR95 ↓ | $63.31^{\pm 4.63}$ | $9.14^{\pm 0.46}$ | $0.12^{\pm 0.07}$ | $0.05^{\pm 0.03}$ | 18.16 |
| $\beta = 1/\sqrt{D}$ | AUROC ↑ | $89.09^{\pm 0.66}$ | $90.59^{\pm 0.35}$ | $99.59^{\pm 0.18}$ | $\mathbf{99.87}^{\pm \mathbf{0.01}}$ | 94.79 |
| | FPR95 ↓ | $53.96^{\pm 3.30}$ | $47.50^{\pm 1.83}$ | $0.17^{\pm 0.18}$ | $0.04^{\pm 0.02}$ | 25.42 |
| | | | Output OOD | | | |
| $\beta = 0$ | AUROC ↑ | $77.67^{\pm 1.37}$ | $85.10^{\pm 0.61}$ | $84.12^{\pm 1.08}$ | $91.70^{\pm 0.44}$ | 84.65 |
| | FPR95 ↓ | $82.07^{\pm 1.30}$ | $69.32^{\pm 1.65}$ | $57.30^{\pm 1.73}$ | $29.37^{\pm 1.73}$ | 59.52 |
| $\beta = 0.25$ | AUROC ↑ | $91.37^{\pm 0.64}$ | $\mathbf{93.66}^{\pm \mathbf{0.13}}$ | $94.79^{\pm 0.29}$ | $96.56^{\pm 0.27}$ | 94.10 |
| | FPR95 ↓ | $43.03^{\pm 1.71}$ | $34.70^{\pm 0.32}$ | $38.38^{\pm 3.27}$ | $18.61^{\pm 2.44}$ | 33.68 |
| $\beta = 0.5$ | AUROC ↑ | $\mathbf{93.37}^{\pm \mathbf{0.54}}$ | $\underline{92.62}^{\pm 0.67}$ | $\mathbf{98.04}^{\pm \mathbf{0.28}}$ | $\mathbf{98.30}^{\pm \mathbf{0.11}}$ | **95.59** |
| | FPR95 ↓ | $23.12^{\pm 1.97}$ | $29.91^{\pm 2.93}$ | $6.34^{\pm 1.56}$ | $6.83^{\pm 0.64}$ | 16.55 |
| $\beta = 1$ | AUROC ↑ | $93.06^{\pm 0.57}$ | $91.82^{\pm 0.71}$ | $\underline{97.66}^{\pm 0.33}$ | $97.91^{\pm 0.22}$ | 95.11 |
| | FPR95 ↓ | $24.04^{\pm 1.95}$ | $32.04^{\pm 2.97}$ | $\underline{9.29}^{\pm 1.71}$ | $8.82^{\pm 1.42}$ | 18.55 |
| $\beta = 2$ | AUROC ↑ | $\underline{93.25}^{\pm 0.48}$ | $91.98^{\pm 0.73}$ | $97.57^{\pm 0.40}$ | $\underline{97.97}^{\pm 0.19}$ | $\underline{95.19}$ |
| | FPR95 ↓ | $\underline{23.69}^{\pm 1.94}$ | $\underline{31.23}^{\pm 3.09}$ | $10.06^{\pm 2.44}$ | $\underline{8.37}^{\pm 1.30}$ | $\underline{18.34}$ |
| $\beta = 1/\sqrt{D}$ | AUROC ↑ | $54.67^{\pm 0.72}$ | $80.59^{\pm 0.72}$ | $94.12^{\pm 0.30}$ | $94.93^{\pm 0.35}$ | 81.08 |
| | FPR95 ↓ | $92.40^{\pm 0.21}$ | $65.83^{\pm 1.03}$ | $30.04^{\pm 1.15}$ | $27.20^{\pm 1.94}$ | 53.87 |

show that $s(\boldsymbol{Z})$ achieves better OOD discrimination w.r.t. the mean AUROC and FPR95. While $s_{\min}(\boldsymbol{Z})$ roughly matches the OOD detection metrics of $s(\boldsymbol{Z})$ on CNN/DM for both input and output, $s_{\min}(\boldsymbol{Z})$ lags behind $s(\boldsymbol{Z})$ on the other OOD data sets.

## D.8 ABLATIONS

**Beta sensitivity analysis.** We evaluate AP-OOD when varying the hyperparameter $\beta$ on the summarization task. We select $\beta$ from $\{0, 1/\sqrt{D}, 0.25, 0.5, 1, 2\}$, and we leave the settings for $M$ and $T$ unchanged (i.e., they are identical to the settings used in Table 1). Table 8 shows that AP-OOD on text summarization is relatively insensitive to the selection of $\beta$ inside the range $[0.25, 2]$ in the input and output settings.

Table 9: Unsupervised OOD detection performance on text summarization. We compare results from AP-OOD trained on XSUM as the ID data set when varying $M$ and $T$. $\downarrow$ indicates "lower is better" and $\uparrow$ "higher is better". All values in %. We estimate standard deviations across five independent dataset splits and training runs.

| | | CNN/DM | Newsroom | Reddit | Samsum | Mean |
|---|---|---|---|---|---|---|
| | | Input OOD | | | | |
| $M=1024$  $T=1$ | AUROC $\uparrow$ | $97.16^{\pm0.22}$ | $98.25^{\pm0.11}$ | $99.82^{\pm0.01}$ | $99.32^{\pm0.03}$ | $98.64$ |
| | FPR95 $\downarrow$ | $14.72^{\pm0.83}$ | $7.54^{\pm0.62}$ | $\mathbf{0.00^{\pm0.00}}$ | $0.64^{\pm0.11}$ | $5.72$ |
| $M=512$  $T=2$ | AUROC $\uparrow$ | $\mathbf{97.98^{\pm0.16}}$ | $\mathbf{98.83^{\pm0.07}}$ | $\underline{99.87^{\pm0.03}}$ | $\mathbf{99.60^{\pm0.04}}$ | $\mathbf{99.07}$ |
| | FPR95 $\downarrow$ | $\mathbf{9.77^{\pm0.80}}$ | $\mathbf{4.67^{\pm0.30}}$ | $\mathbf{0.00^{\pm0.00}}$ | $0.02^{\pm0.02}$ | $\mathbf{3.61}$ |
| $M=256$  $T=4$ | AUROC $\uparrow$ | $97.76^{\pm0.11}$ | $98.75^{\pm0.07}$ | $\mathbf{99.87^{\pm0.06}}$ | $99.46^{\pm0.09}$ | $98.96$ |
| | FPR95 $\downarrow$ | $11.07^{\pm0.74}$ | $\underline{4.75^{\pm0.41}}$ | $\mathbf{0.00^{\pm0.00}}$ | $\underline{0.02^{\pm0.02}}$ | $\underline{3.96}$ |
| $M=128$  $T=8$ | AUROC $\uparrow$ | $97.53^{\pm0.15}$ | $98.49^{\pm0.15}$ | $99.83^{\pm0.07}$ | $99.14^{\pm0.12}$ | $98.75$ |
| | FPR95 $\downarrow$ | $12.48^{\pm1.14}$ | $5.94^{\pm0.65}$ | $\mathbf{0.00^{\pm0.00}}$ | $0.25^{\pm0.10}$ | $4.67$ |
| $M=64$  $T=16$ | AUROC $\uparrow$ | $97.10^{\pm0.09}$ | $98.14^{\pm0.16}$ | $99.84^{\pm0.07}$ | $98.81^{\pm0.16}$ | $98.47$ |
| | FPR95 $\downarrow$ | $14.30^{\pm0.77}$ | $7.87^{\pm0.86}$ | $0.00^{\pm0.00}$ | $0.99^{\pm0.50}$ | $5.79$ |
| $M=32$  $T=32$ | AUROC $\uparrow$ | $96.84^{\pm0.35}$ | $97.78^{\pm0.15}$ | $99.83^{\pm0.05}$ | $98.56^{\pm0.28}$ | $98.25$ |
| | FPR95 $\downarrow$ | $14.97^{\pm1.96}$ | $10.18^{\pm0.80}$ | $0.01^{\pm0.02}$ | $2.53^{\pm2.12}$ | $6.92$ |
| $M=16$  $T=64$ | AUROC $\uparrow$ | $96.23^{\pm0.45}$ | $97.35^{\pm0.24}$ | $99.73^{\pm0.11}$ | $98.12^{\pm0.24}$ | $97.86$ |
| | FPR95 $\downarrow$ | $16.65^{\pm1.99}$ | $12.55^{\pm1.15}$ | $0.09^{\pm0.20}$ | $5.69^{\pm1.87}$ | $8.75$ |
| $M=8$  $T=128$ | AUROC $\uparrow$ | $95.56^{\pm0.38}$ | $96.47^{\pm0.46}$ | $99.67^{\pm0.27}$ | $97.44^{\pm0.25}$ | $97.29$ |
| | FPR95 $\downarrow$ | $18.16^{\pm1.57}$ | $16.34^{\pm1.91}$ | $0.52^{\pm1.13}$ | $11.29^{\pm1.78}$ | $11.58$ |
| $M=4$  $T=256$ | AUROC $\uparrow$ | $94.58^{\pm0.67}$ | $94.75^{\pm0.52}$ | $99.27^{\pm0.86}$ | $95.24^{\pm0.25}$ | $95.96$ |
| | FPR95 $\downarrow$ | $20.10^{\pm2.32}$ | $21.71^{\pm2.30}$ | $2.01^{\pm4.09}$ | $24.58^{\pm1.83}$ | $17.10$ |
| $M=2$  $T=512$ | AUROC $\uparrow$ | $93.17^{\pm0.75}$ | $91.87^{\pm0.56}$ | $98.43^{\pm2.39}$ | $89.87^{\pm0.86}$ | $93.34$ |
| | FPR95 $\downarrow$ | $22.86^{\pm2.20}$ | $27.09^{\pm1.48}$ | $4.95^{\pm9.38}$ | $39.75^{\pm3.06}$ | $23.66$ |
| $M=1$  $T=1024$ | AUROC $\uparrow$ | $90.90^{\pm1.20}$ | $88.10^{\pm0.83}$ | $96.68^{\pm5.76}$ | $81.41^{\pm1.06}$ | $89.27$ |
| | FPR95 $\downarrow$ | $27.14^{\pm3.03}$ | $32.64^{\pm2.29}$ | $9.03^{\pm16.78}$ | $52.73^{\pm3.76}$ | $30.39$ |
| | | Output OOD | | | | |
| $M=1024$  $T=1$ | AUROC $\uparrow$ | $92.47^{\pm0.48}$ | $94.17^{\pm0.30}$ | $98.36^{\pm0.22}$ | $97.77^{\pm0.14}$ | $95.69$ |
| | FPR95 $\downarrow$ | $39.11^{\pm1.81}$ | $34.69^{\pm0.85}$ | $3.11^{\pm1.16}$ | $12.59^{\pm0.90}$ | $22.38$ |
| $M=512$  $T=2$ | AUROC $\uparrow$ | $\mathbf{93.79^{\pm0.25}}$ | $\mathbf{95.85^{\pm0.18}}$ | $99.02^{\pm0.20}$ | $98.96^{\pm0.06}$ | $\mathbf{96.90}$ |
| | FPR95 $\downarrow$ | $\mathbf{32.45^{\pm1.29}}$ | $\mathbf{20.10^{\pm0.67}}$ | $0.95^{\pm0.66}$ | $2.77^{\pm0.54}$ | $\mathbf{14.07}$ |
| $M=256$  $T=4$ | AUROC $\uparrow$ | $93.35^{\pm0.46}$ | $\underline{95.48^{\pm0.28}}$ | $99.19^{\pm0.26}$ | $\mathbf{99.05^{\pm0.06}}$ | $\underline{96.77}$ |
| | FPR95 $\downarrow$ | $33.67^{\pm2.77}$ | $21.73^{\pm0.82}$ | $0.86^{\pm0.95}$ | $2.72^{\pm0.52}$ | $\underline{14.75}$ |
| $M=128$  $T=8$ | AUROC $\uparrow$ | $93.24^{\pm0.34}$ | $95.27^{\pm0.37}$ | $\mathbf{99.21^{\pm0.41}}$ | $\underline{98.99^{\pm0.04}}$ | $96.68$ |
| | FPR95 $\downarrow$ | $\underline{32.84^{\pm1.75}}$ | $23.40^{\pm1.53}$ | $0.99^{\pm1.56}$ | $3.26^{\pm0.42}$ | $15.12$ |
| $M=64$  $T=16$ | AUROC $\uparrow$ | $92.95^{\pm0.82}$ | $94.92^{\pm0.39}$ | $99.11^{\pm0.36}$ | $98.89^{\pm0.14}$ | $96.47$ |
| | FPR95 $\downarrow$ | $34.08^{\pm4.42}$ | $25.53^{\pm1.87}$ | $1.48^{\pm1.63}$ | $4.10^{\pm0.70}$ | $16.30$ |
| $M=32$  $T=32$ | AUROC $\uparrow$ | $92.54^{\pm0.61}$ | $94.11^{\pm0.47}$ | $98.67^{\pm0.73}$ | $98.63^{\pm0.41}$ | $95.99$ |
| | FPR95 $\downarrow$ | $37.21^{\pm3.76}$ | $29.56^{\pm2.71}$ | $4.68^{\pm4.39}$ | $6.11^{\pm2.55}$ | $19.39$ |
| $M=16$  $T=64$ | AUROC $\uparrow$ | $91.26^{\pm1.17}$ | $92.62^{\pm1.40}$ | $97.99^{\pm2.33}$ | $98.58^{\pm0.84}$ | $95.11$ |
| | FPR95 $\downarrow$ | $41.96^{\pm4.43}$ | $35.78^{\pm5.78}$ | $8.75^{\pm13.44}$ | $6.19^{\pm4.88}$ | $23.17$ |
| $M=8$  $T=128$ | AUROC $\uparrow$ | $90.94^{\pm1.97}$ | $91.99^{\pm1.88}$ | $97.10^{\pm2.54}$ | $98.28^{\pm0.80}$ | $94.58$ |
| | FPR95 $\downarrow$ | $41.24^{\pm8.00}$ | $36.42^{\pm7.58}$ | $13.13^{\pm13.35}$ | $7.58^{\pm3.85}$ | $24.59$ |
| $M=4$  $T=256$ | AUROC $\uparrow$ | $89.62^{\pm1.80}$ | $90.35^{\pm2.64}$ | $95.91^{\pm3.26}$ | $97.73^{\pm0.96}$ | $93.40$ |
| | FPR95 $\downarrow$ | $47.52^{\pm9.04}$ | $41.77^{\pm12.21}$ | $18.53^{\pm16.24}$ | $10.02^{\pm4.76}$ | $29.46$ |
| $M=2$  $T=512$ | AUROC $\uparrow$ | $87.82^{\pm2.50}$ | $88.06^{\pm1.29}$ | $94.00^{\pm3.38}$ | $96.91^{\pm1.26}$ | $91.70$ |
| | FPR95 $\downarrow$ | $52.18^{\pm9.71}$ | $50.66^{\pm5.51}$ | $28.44^{\pm17.40}$ | $13.98^{\pm6.18}$ | $36.31$ |
| $M=1$  $T=1024$ | AUROC $\uparrow$ | $86.45^{\pm1.86}$ | $86.95^{\pm1.79}$ | $93.43^{\pm2.35}$ | $96.10^{\pm1.59}$ | $90.73$ |
| | FPR95 $\downarrow$ | $50.92^{\pm8.94}$ | $49.61^{\pm6.70}$ | $29.61^{\pm8.37}$ | $14.82^{\pm3.62}$ | $36.24$ |

**Number of heads $M$ and queries $T$.** We ablate on the number of heads $M$ and the number of queries $T$ of AP-OOD on the summarization task. For this ablation, we select $T \in \{1, 2, 4, 8, 16, 32, 64, 128, 512, 1024\}$ and we then select $M$ such that the total number of parameters of AP-OOD equals the number of entries in $\mathbf{\Sigma}$ of the Mahalanobis method, i.e., such that $MT = D$. The results in Table 9 show that AP-OOD works best on the summarization task for both input and output when $M = 512$ and $T = 2$. Although the performance drops when decreasing $M$ and increasing $T$, we find that AP-OOD is relatively insensitive to the number of heads and queries.

**Dot product and Euclidean distance.** We compare using the dot product and the negative squared Euclidean distance for the attention pooling in AP-OOD. For a formal definition of attention pooling with the negative squared Euclidean distance, we refer to Appendix D.1. Table 10 shows that using the dot product works substantially better. This result aligns with the well-established observation that measuring similarity using the dot product in high-dimensional spaces is more effective than using Euclidean distance.

Table 10: Unsupervised OOD detection performance on text summarization. We compare results from AP-OOD trained on XSUM as the ID data set when using the dot product and the Euclidean similarity. ↓ indicates "lower is better" and ↑ "higher is better". All values in %. We estimate standard deviations across five independent dataset splits and training runs.

| | | CNN/DM | Newsroom | Reddit | Samsum | Mean |
|---|---|---|---|---|---|---|
| | | Input OOD | | | | |
| Dot product | AUROC ↑ | $97.76^{\pm 0.11}$ | $98.75^{\pm 0.07}$ | $99.87^{\pm 0.06}$ | $99.46^{\pm 0.09}$ | **98.96** |
| | FPR95 ↓ | $11.07^{\pm 0.74}$ | $4.75^{\pm 0.41}$ | $0.00^{\pm 0.00}$ | $0.02^{\pm 0.02}$ | **3.96** |
| Euclidean | AUROC ↑ | $74.22^{\pm 0.65}$ | $84.43^{\pm 0.23}$ | $97.06^{\pm 0.41}$ | $98.30^{\pm 0.23}$ | 88.50 |
| | FPR95 ↓ | $90.20^{\pm 0.37}$ | $74.08^{\pm 1.04}$ | $15.27^{\pm 5.30}$ | $7.17^{\pm 1.94}$ | 46.68 |
| | | Output OOD | | | | |
| Dot product | AUROC ↑ | $93.37^{\pm 0.54}$ | $92.62^{\pm 0.65}$ | $98.04^{\pm 0.29}$ | $98.30^{\pm 0.11}$ | **95.58** |
| | FPR95 ↓ | $23.12^{\pm 1.98}$ | $29.93^{\pm 2.89}$ | $6.36^{\pm 1.60}$ | $6.83^{\pm 0.64}$ | **16.56** |
| Euclidean | AUROC ↑ | $87.67^{\pm 0.74}$ | $88.17^{\pm 1.80}$ | $96.50^{\pm 0.57}$ | $91.28^{\pm 1.79}$ | 90.90 |
| | FPR95 ↓ | $65.62^{\pm 3.90}$ | $66.04^{\pm 4.38}$ | $22.34^{\pm 5.36}$ | $53.89^{\pm 7.80}$ | 51.97 |

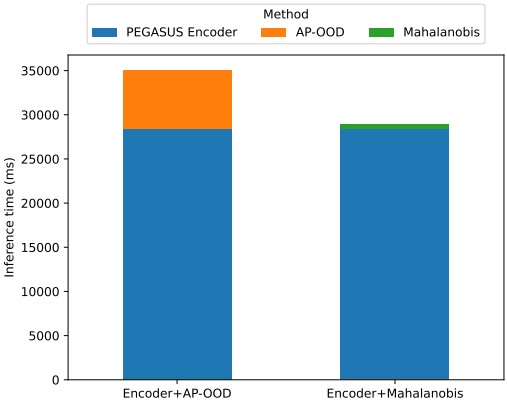

Figure 7: Comparing AP-OOD and the Mahalanobis method relative to the encoder inference time. The bars show the mean over ten batches.

## D.9 PERFORMANCE MEASUREMENTS

We analyze the inference time of AP-OOD in comparison to the transformer backbone and other OOD detection methods. To avoid bottlenecks during data loading, we measure inference times on single batches and report the mean and standard deviation across 10 batches. Our measurements only start after a warm-up phase of 5 batches. If not stated otherwise, the measurements were performed with a batch size of 32 and context length of 512 tokens. All measurements were performed on a single NVIDIA A100-40GB GPU.

Figure 8 compares the inference time of various OOD detection methods for different batch sizes. As expected AP-OOD has a strong linear relation to the batch size and is significantly slower than the reference models.

Although AP-OOD is slower than other methods like the Mahalanobis method, Figure 7 illustrates that it still takes less than 20 % of the combined inference time of the PEGASUS

Table 11: Inference times of OOD methods and PEGASUS transformer for a batch size of 32 samples, number of heads $M = 256$, number of queries $T = 4$, and a context length of $S = 512$ tokens. All values in milliseconds $ms$. We estimate the mean and standard deviation over ten batches.

| AP-OOD | Mahalanobis | PEGASUS Encoder | PEGASUS Generation |
|---|---|---|---|
| $6.58^{\pm 0.095}$ | $0.52^{\pm 0.146}$ | $28.43^{\pm 2.513}$ | $34940.34^{\pm 65.842}$ |

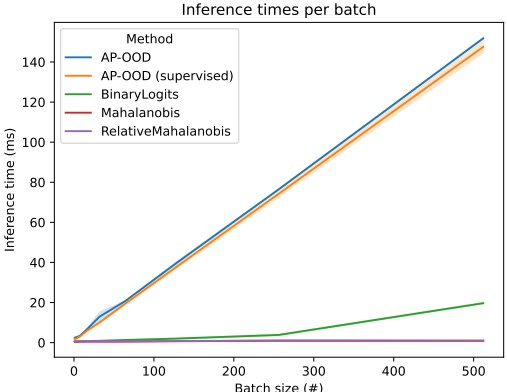

Figure 8: Comparison of various OOD detection methods for increasing batch sizes. We estimate the mean and standard deviation over ten batches.

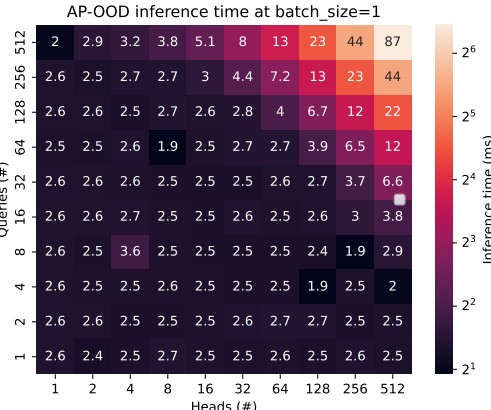

Figure 9: Inference times for AP-OOD over different numbers of heads $M$ and queries $T$. We estimate the mean over ten batches.

encoder and OOD detection method. We argue that the higher OOD detection rate mitigates the addition of overhead of AP-OOD since it allows skipping the substantially longer generation time more often (see Table 11). The degree to which the overhead of AP-OOD is mitigated by skipping the decoder depends on the rate of detected OOD samples in future applications.

The heatmap in Figure 9 shows the inference time of AP-OOD for different selections of the hyperparameter number of heads $M$ and number of queries $T$. While for small values of both parameters the inference time is constant, for larger parameters the inference time increases linearly with both parameters.

The inference time of the decoder of the transformer depends on the length of the longest output sequence of a batch. To obtain consistent measurements, we forced the decoder to always produce the same sequence length. Figure 10 illustrates the inference times of the PEGASUS transformer model. The plots of the first row cover the performance of the PEGASUS encoder only, while the second row shows the combined inference time of the encoder and decoder. In the left column, the plots indicate that the model inference time increases linearly with the batch size. Further, it is shown that the encoder takes about 10 % of the overall inference time. The right column shows the inference times for an increasing number of context tokens. The inference time of the transformer encoder and decoder increases quadratically with the context length.

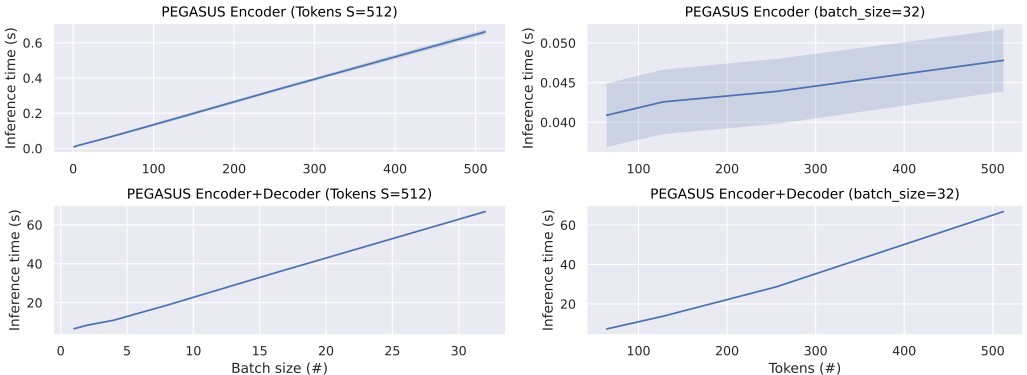

Figure 10: Inference times for the PEGASUS model for various batch sizes (left) and various numbers of input tokens (right). The top row illustrates the Encoder's performance, while the bottom row shows the combined performance of the Encoder and Decoder. We estimate the mean and standard deviation over ten batches.

## E    DETAILS ON CONTINUOUS MODERN HOPFIELD NETWORKS

The following arguments are adopted from Hofmann et al. (2024); Fürst et al. (2022) and Ramsauer et al. (2021). Associative memory networks have been designed to store and retrieve samples. Hopfield networks are energy-based, binary associative memories, which were popularized as artificial neural network architectures in the 1980s (Hopfield, 1982; 1984). Their storage capacity can be considerably increased by polynomial terms in the energy function (Chen et al., 1986; Psaltis & Park, 1986; Baldi & Venkatesh, 1987; Gardner, 1987; Abbott & Arian, 1987; Horn & Usher, 1988; Caputo & Niemann, 2002; Krotov & Hopfield, 2016). In contrast to these binary memory networks, we use continuous associative memory networks with far higher storage capacity. These networks are continuous and differentiable, retrieve with a single update, and have exponential storage capacity (and are therefore scalable, i.e., able to tackle large problems; Ramsauer et al., 2021).

Formally, we denote a set of patterns $\{\boldsymbol{x}_1, \ldots, \boldsymbol{x}_N\} \subset \mathbb{R}^d$ that are stacked as columns to the matrix $\boldsymbol{X} = (\boldsymbol{x}_1, \ldots, \boldsymbol{x}_N)$ and a state pattern (query) $\boldsymbol{\xi} \in \mathbb{R}^d$ that represents the current state. The largest norm of a stored pattern is $M = \max_i \|\boldsymbol{x}_i\|$. Then, the energy E of continuous Modern Hopfield Networks with state $\boldsymbol{\xi}$ is defined as (Ramsauer et al., 2021)

$$\text{E} = -\beta^{-1} \log\left(\sum_{i=1}^N \exp(\beta \boldsymbol{x}_i^T \boldsymbol{\xi})\right) + \frac{1}{2}\boldsymbol{\xi}^T\boldsymbol{\xi} + \text{C}, \tag{56}$$

where $\text{C} = \beta^{-1}\log N + \frac{1}{2}M^2$. For energy E and state $\boldsymbol{\xi}$, Ramsauer et al. (2021) proved that the update rule

$$\boldsymbol{\xi}^{\text{new}} = \boldsymbol{X}\,\text{softmax}(\beta\boldsymbol{X}^T\boldsymbol{\xi}) \tag{57}$$

converges globally to stationary points of the energy E and coincides with the attention mechanisms of Transformers (Vaswani et al., 2017a; Ramsauer et al., 2021).

The *separation* $\Delta_i$ of a pattern $\boldsymbol{x}_i$ is its minimal dot product difference to any of the other patterns:

$$\Delta_i = \min_{j, j \neq i}\left(\boldsymbol{x}_i^T\boldsymbol{x}_i - \boldsymbol{x}_i^T\boldsymbol{x}_j\right). \tag{58}$$

A pattern is *well-separated* from the data if $\Delta_i$ is above a given threshold (specified in Ramsauer et al., 2021). If the patterns $\boldsymbol{x}_i$ are well-separated, the update rule in Equation (57) converges to a fixed point close to a stored pattern. If some patterns are similar to one another and, therefore, not well-separated, the update rule converges to a fixed point close to the mean of the similar patterns.

The update rule of a Hopfield network thus identifies sample–sample relations between stored patterns. This enables similarity-based learning methods like nearest neighbor search (see Schäfl et al., 2022).

Hopfield networks have recently been used for OOD detection (Zhang et al., 2023b; Hofmann et al., 2024). Hu et al. (2024) introduces Hopfield layers for outlier-efficient memory update.

## F    THE USE OF LARGE LANGUAGE MODELS

When creating this paper, we utilized large language models (LLMs) to refine our writing, to identify related work, and for research ideation. When refining the writing using LLMs, we carefully review and verify LLM output to preserve sentence semantics. For related work, we confirm the soundness of papers suggested by the LLM, and for research ideation, we verify the factual accuracy of all statements.

