# OpenReview forum: "AP-OOD: Attention Pooling for Out-of-Distribution Detection"
_ICLR.cc/2026/Conference — ICLR 2026 Poster_

### Official Review · Reviewer_QF7M · 2025-10-28

**Soundness:** 3
**Presentation:** 2
**Contribution:** 2
**Rating:** 4
**Confidence:** 4

**Summary:**

This paper proposes AP-OOD, an iteration on OOD detection using Mahalanobis distance with token embeddings [1]. In particular, by replaces mean pooling of token embeddings with learnable attention pooling to improve OOD detection in text models. The authors provide rigorous theoretical background as well as empirical evaluations to showcase the validity of AP-OOD. Additionally, the paper presents the methodology in both its supervised formulation (leveraging auxiliary data) and unsupervised formulation.

[1] Jie Ren, Jiaming Luo, Yao Zhao, Kundan Krishna, Mohammad Saleh, Balaji Lakshminarayanan, and Peter J Liu. Out-of-distribution detection and selective generation for conditional language models. In The Eleventh International Conference on Learning Representations, 2023.

**Strengths:**

The reviewer notes the following strengths:
- The paper presents a very clear motivation, with clear justifications on why its nessacary to leverage token-level information beyond mean-pooling.
- The paper is backed with sound theoretical background which helps frame the entire work and adds valuable contribution to the OOD detection field.
- The empirical analysis showcases meaningful improvements upon the baseline, with indications that the methodology can be applicable beyond the base text modality (given audio classification evaluations).

**Weaknesses:**

The reviewer notes the following weakness:
- Although the improvement is measurable, the incorporation of learnable attention pooling into Mahalanobis distance is not conceptually groundbreaking and may hinder the novelty of the overall work.
- The paper presents few intuitive explanations for overall observations leading to difficulties in finding the novelty of the work.
- The lack of experiments on larger, more realistic language models, potentially makes the work less applicable in real-world settings.

**Questions:**

The reviewer would like to encourage the authors to revisit the presentation of the overall work. In particular, much of the background can be attributed to Ren et al's paper and could be instead used to visit some of the ablations experiments presented in the appendix [1]. Additionally, more focus on the novelty of the underlying work whether that be the theoretical or algorithmic contribution would help improve the overall quality of the paper.

[1] Jie Ren, Jiaming Luo, Yao Zhao, Kundan Krishna, Mohammad Saleh, Balaji Lakshminarayanan, and Peter J Liu. Out-of-distribution detection and selective generation for conditional language models. In The Eleventh International Conference on Learning Representations, 2023.

---

> ### Author Response · Authors · 2025-11-22
>
> **Response to Weaknesses**
>
> > Although the improvement is measurable, the incorporation of learnable attention pooling into Mahalanobis distance is not conceptually groundbreaking and may hinder the novelty of the overall work. The paper presents few intuitive explanations for overall observations leading to difficulties in finding the novelty of the work.
>
> We respectfully disagree with the assessment that AP-OOD’s novelty is limited. Our main goal when writing the manuscript was to provide a clear motivation for AP-OOD by naturally connecting the baseline from Ren et al. (2023) to AP-OOD. This might have obscured the novelty of our work. Incorporating attention pooling by itself is a non-trivial novelty. Our core contribution is the unique integration of attention pooling with the Mahalanobis distance. This is a significant step, as the standard formulation makes this impossible. Our approach critically depends on leveraging a directional decomposition to enable this novel combination. Another novelty we contribute is to use the directional decomposition with attention pooling to derive an OOD score as defined in Equation (2) in the manuscript. The third core novelty is the extension of standard attention pooling to use multiple queries per head. In conventional attention pooling, the softmax function operates over a single dimension, normalizing only across the sequence of keys. Our extension, however, introduces a matrix-valued softmax. This novel function is normalized jointly over the entire $S \times T$ interaction space—that is, over both the $S$ sequence tokens (keys) and the $T$ query weight vectors. We have revised the manuscript to underscore the novelty of our approach.
>
> > The lack of experiments on larger, more realistic language models, potentially makes the work less applicable in real-world settings.
>
> We received similar feedback from reviewer `18Vp`. We see three main challenges in applying the setting  from Ren et al. (2023) to modern LLMs:
> 1. The setting assumes access to the model’s training data for fitting the OOD detector. However, the training data sets for many LLMs (including the three LLMs mentioned by the reviewer) are not publicly available.
>
> 2. Many LLMs are trained on large amounts of text. This makes it difficult to define appropriate data sets that are fully OOD w.r.t. the training data for evaluation. Finding a large and diverse auxiliary outlier data set for training an OOD detector in the supervised setting is even more challenging.
>
> 3. The training process of LLMs usually involves multiple stages (pre-training, instruction tuning, RLHF, …), making it difficult to pinpoint an exact training distribution $p_\text{ID}$ for an LLM.
>
> For these reasons, we consider OOD detection for modern LLM pipelines as outside the scope of our work.
>
> However, we have included a discussion regarding LLMs in the Future Work section of the updated manuscript. And, to verify the general applicability of AP-OOD to the decoder-only language modeling paradigm of LLMs, in addition to the existing summarization and translation tasks, we now conduct experiments on Pythia-160M (a decoder-only LM trained on the Pile, a large open-source data set of English text). We have included the experiments and results in Appendix D.6 in the updated manuscript. To summarize, we compare OOD detection results across five OOD test data sets and find that AP-OOD outperforms the second-best method (Perplexity) in terms of both mean AUROC (from 77.59% to 86.67%) and mean FPR95 (from 59.25% to 55.81%).
>
> **Response to Questions**
>
> > The reviewer would like to encourage the authors to revisit the presentation of the overall work. In particular, much of the background can be attributed to Ren et al's paper and could be instead used to visit some of the ablations experiments presented in the appendix [1]. Additionally, more focus on the novelty of the underlying work whether that be the theoretical or algorithmic contribution would help improve the overall quality of the paper.
>
> We have uploaded an updated version of the manuscript with the following improvements: We extended the introduction to connect the problem of language model hallucination to OOD inputs and to describe the limitations of existing seminal OOD detection approaches in language modeling, we revised the Method section and stressed novelty more, we have included  PyTorch/Einops-like pseudocode implementing the loss computation of AP-OOD in Appendix C.2, and we have included a discussion regarding LLMs in the future work section of the updated manuscript.
>
> We also implemented the reviewer’s feedback on additional ablations in the main body of the manuscript. Therefore, we included a new scalability study for AP-OOD. In this study, we no longer restrict AP-OOD’s parameter count to match the Mahalanobis baseline. The results (Figure 4) show that AP-OOD’s OOD detection performance increases with both the number of heads ($M$) and the number of queries ($T$).

---

> ### Author Response · Authors · 2025-11-22
>
> **References**
>
> Ren, Jie, et al. "Out-of-distribution detection and selective generation for conditional language models." In ICLR 2023.

---

### Official Review · Reviewer_M9r8 · 2025-10-28

**Soundness:** 1
**Presentation:** 1
**Contribution:** 2
**Rating:** 2
**Confidence:** 3

**Summary:**

This paper introduces AP-OOD, a novel method for out-of-distribution (OOD) detection in natural language tasks that leverages token-level information instead of simple average-based embedding aggregation. Operating in a semi-supervised framework, AP-OOD flexibly combines unsupervised and supervised approaches by using limited auxiliary outlier data.

**Strengths:**

- The motivation for using standard Gaussian-based averaging is well explained.
- The paper presents an interesting and original idea.
- The authors thoughtfully explore multiple use cases to demonstrate the method’s applicability.

**Weaknesses:**

- The paper suffers from clarity and presentation issues. The introduction does not sufficiently articulate the limitations of existing methods or explain how the proposed approach addresses them. The methodology section is overly condensed and lacks precision, with several symbols insufficiently defined. Moreover, the authors do not reference the accompanying pseudocode. Overall, the manuscript appears hastily written and insufficiently polished.

- Although the approach is presented as semi-supervised, the paper merely demonstrates a transition between supervised and unsupervised scenarios without providing a clear formulation or justification for the semi-supervised setting. Additionally, the proposed supervised out-of-distribution scenario appears limited in practical applicability, as it does not reflect realistic deployment conditions.

- The experimental setup is restricted to a simplistic baseline where embeddings are averaged across sequences, effectively assuming a Gaussian distribution. This assumption is unnecessarily restrictive, as the embedding distribution could be more accurately modeled using mixture models or more expressive probabilistic frameworks such as normalizing flows.

- The concatenation of Z-embeddings (Algorithm 1, line 8) is likely infeasible for larger datasets due to significant memory constraints, which raises concerns about the scalability of the proposed approach.

- The proposed model assumes access to the training data during OOD detection, which severely limits its applicability in realistic or privacy-sensitive scenarios where such access is unavailable.

- The experimental comparison is incomplete. The authors evaluate primarily against standard baselines while omitting several important domain-specific OOD detection methods, such as:

Directed Sparsification – Yiyou Sun and Yixuan Li (2022), DICE: Leveraging Sparsification for Out-of-Distribution Detection.

Virtual-logit Matching – Haoqi Wang, Zhizhong Li, Litong Feng, and Wayne Zhang (2022), VIM: Out-of-Distribution Detection with Virtual-Logit Matching (NeurIPS).

GradNorm – Rui Huang, Andrew Geng, and Yixuan Li (2021), On the Importance of Gradients for Detecting Distributional Shifts in the Wild (NeurIPS).

**Questions:**

Please refer to the weakness section.

---

> ### Author Response · Authors · 2025-11-22
>
> **Response to Weaknesses**
>
> > The paper suffers from clarity and presentation issues. The introduction does not sufficiently articulate the limitations of existing methods or explain how the proposed approach addresses them. The methodology section is overly condensed and lacks precision, with several symbols insufficiently defined. Moreover, the authors do not reference the accompanying pseudocode. Overall, the manuscript appears hastily written and insufficiently polished.
>
> To follow a more common structure with a summarized problem description, we have now added a paragraph that clearly states the limitations of existing work by moving this information from the paragraph on OOD detection in the related work section to the introduction. It reads as follows:
>
> *Many existing post-hoc OOD detection methods assume a classifier as the base model. In contrast, in language modeling, the base model is typically an autoregressive generative model without an explicit classification head. This necessitates the development of OOD detection methods specifically tailored for language modeling, and we believe that the OOD detection community can benefit from generative language modeling as an additional benchmark.*
>
> We acknowledge that the methodology section is dense. Our goal is to outline the complete framework for both unsupervised and semi-supervised settings within the limited space using mathematical notation. To complement the mathematical definitions with a practical perspective, we have included the loss computation of AP-OOD in PyTorch/Einops-like pseudocode in Appendix C.2.
>
> In response to your feedback regarding symbol definitions, we have conducted a thorough review of the manuscript. We verified that all mathematical notation is now explicitly defined, and we have added supplementary context to specific symbols to ensure clarity.
>
> Indeed, we did miss referencing Algorithm 1 in the main text of our original submission. We have addressed this issue and added the reference in the updated manuscript.
>
> > Although the approach is presented as semi-supervised, the paper merely demonstrates a transition between supervised and unsupervised scenarios without providing a clear formulation or justification for the semi-supervised setting. Additionally, the proposed supervised out-of-distribution scenario appears limited in practical applicability, as it does not reflect realistic deployment conditions.
>
> We agree that this is a presentation issue in our work. To resolve this issue, we added references that address this widely used problem setting in the updated manuscript. To us, the semi-supervised setting is both interesting and practically relevant (Yoon et al., 2023; Qiao et al., 2024; Ivanov et al., 2024). In many cases, practitioners only have access to a small or restricted auxiliary outlier data set. For example, consider OOD detection on a translation task that translates from a less widely spoken source language. As another example, consider detecting defects in industrial machines using recordings of their sounds (Nishida et al., 2025). Practitioners can collect a relatively large amount of ID audio data from machines while they run without defects. However, it is much harder to collect diverse auxiliary outlier examples from defective machines because defects are infrequent and different defects can affect audio recordings in distinct ways. We have expanded the motivation on the semi-supervised setting in the updated version of the manuscript to include the defect detection task as an additional, practically relevant example.
>
> Of course, there also exist many settings that permit large and diverse auxiliary outlier data sets for training the OOD detector. In this fully supervised setting, we show in Appendix D.3 and D.5 that AP-OOD achieves state-of-the-art OOD detection results on both summarization and translation.

---

> ### Author Response · Authors · 2025-11-22
>
> > The experimental setup is restricted to a simplistic baseline where embeddings are averaged across sequences, effectively assuming a Gaussian distribution. This assumption is unnecessarily restrictive, as the embedding distribution could be more accurately modeled using mixture models or more expressive probabilistic frameworks such as normalizing flows.
>
> We fully agree with the reviewer’s observation that the mean-pooling-based approaches from Ren et al. (2023) are limited. By proposing AP-OOD, we aim to address exactly this limitation. We also agree that more complex approaches might also solve this issue; however, there is also a risk that they may not work when tried. Both of the approaches the reviewer suggested are highly non-trivial to apply to OOD detection on tokenized data:
>
> - Mixture models: It is possible to fit a mixture model on a single sequence of token embeddings. However, when a sequence is represented by a mixture model (rather than a vector), determining whether this mixture model (and therefore the sequence) is ID or OOD is highly non-trivial. While compelling, representing sequences as distributions rather than vectors complicates comparisons. Standard metrics like the Mahalanobis distance are not directly applicable in this context. Instead, we would likely need to rely on distributional metrics, such as the Wasserstein distance or the KL-divergence. This fundamentally reframes the task from outlier detection in a vector space to the more complex challenge of detecting outliers within a space of probability distributions.
>
> - Normalizing flows: A key requirement of normalizing flows is that all mappings be invertible. This is overly restrictive for OOD detection (because generating samples is not needed). For example, mean pooling and attention pooling are non-invertible and therefore cannot be used in normalizing flows. Additionally, normalizing flows have been shown to struggle in OOD detection settings (Nalisnick et al., 2019).
>
> > The concatenation of Z-embeddings (Algorithm 1, line 8) is likely infeasible for larger datasets due to significant memory constraints, which raises concerns about the scalability of the proposed approach.
>
> We have indeed considered this problem while conceiving AP-OOD. Our key insight is that it is possible to perform attention pooling over a long sequence by iterating over mini-batches of the sequence. We mentioned this in the "Training" paragraph of the "Experiments" section and included the algorithm for mini-batch attention pooling in the Appendix of the original manuscript. That said, we should have made this more explicit in the paper to avoid misunderstandings. Therefore, we now reference the mini-batch attention pooling process in Algorithm 1 and have added a paragraph describing it in the "Method" section.
>
> > The proposed model assumes access to the training data during OOD detection, which severely limits its applicability in realistic or privacy-sensitive scenarios where such access is unavailable.
>
> It is true that AP-OOD requires access to the training data for learning the OOD detector. However, after AP-OOD is trained (i.e., after the mini-batch attention pooling process), it relies solely on the summary statistics it learned from the data and does not require access to the ID data set to perform OOD detection.
>
> >The experimental comparison is incomplete. The authors evaluate primarily against standard baselines while omitting several important domain-specific OOD detection methods, such as …
>
> We follow Ren et al. (2023) in selecting our baselines. Specifically, we adopt Mahalanobis, relative Mahalanobis, Binary logits, and Perplexity from them. We additionally include Entropy (Malinin et al., 2021), KNN (Sun et al., 2022), Deep SVDD (Ruff et al., 2018), and Deep SAD (Ruff et al., 2019) in our comparison. The methods cited (DICE, ViM, and GradNorm) are important approaches for OOD detection. However, their underlying mechanisms are architecturally specific to discriminative classifiers. Our work addresses OOD detection for auto-regressive language models, which are generative and sequential. This is a distinct problem setting with different challenges. Extending classifier-based OOD detection methods to this setting is not straightforward. We included the arguments above in the introduction, and we also cited the methods the reviewer proposed.

---

> ### Author Response · Authors · 2025-11-22
>
> **References**
>
> Ruff, Lukas, et al. "Deep one-class classification." In ICML 2018.
>
> Ruff, Lukas, et al. "Deep Semi-Supervised Anomaly Detection." In ICLR 2019.
>
> Nalisnick, Eric, et al. "Do deep generative models know what they don't know?." In ICLR 2019.
>
> Malinin, Andrey et al. "Uncertainty Estimation in Autoregressive Structured Prediction." In ICLR 2021.
>
> Sun, Yiyou, et al. "Out-of-distribution detection with deep nearest neighbors." In ICML 2022.
>
> Ren, Jie, et al. "Out-of-distribution detection and selective generation for conditional language models." In ICLR 2023.
>
> Yoon, Jinsung, et al. "SPADE: Semi-supervised Anomaly Detection under Distribution Mismatch." In TMLR 2023.
>
> Ivanov, Viktor et al. "Deep temporal semi-supervised one-class classification for GNSS radio frequency interference detection." In The Journal of Navigation 77.1 (2024): 59-81.
>
> Qiao, Hezhe, et al. "Generative semi-supervised graph anomaly detection." In NeurIPS 2024.
>
> Nishida, Tomoya, et al. “Description and Discussion on DCASE 2025 Challenge Task 2: First-shot Unsupervised Anomalous Sound Detection for Machine Condition Monitoring”. In DCASE workshop 2025.

---

### Official Review · Reviewer_WMzy · 2025-11-01

**Soundness:** 3
**Presentation:** 3
**Contribution:** 2
**Rating:** 6
**Confidence:** 3

**Summary:**

The authors address the problem of out-of-distribution (OOD) detection in natural language processing, highlighting the limitation of mean pooling approaches that lose token-level information. To overcome this, the authors propose AP-OOD, a novel OOD detection method that uses attention pooling to assign higher weights to informative tokens within a sequence. The method builds upon the Mahalanobis distance and introduces attention pooling to better utilize token-level information during OOD scoring. AP-OOD can operate in unsupervised, semi-supervised, and supervised settings and demonstrates improvements over existing baselines across various tasks, including summarization, translation, and audio data.

**Strengths:**

1. This paper moves beyond prior OOD detection approaches that focused on summarizing the entire sentence through mean pooling, introducing a new method that leverages token-level information via attention pooling.

2. By using a toy experiment to intuitively illustrate why attention pooling outperforms mean pooling, the paper clearly and convincingly conveys the core idea of the proposed method.

**Weaknesses:**

1. The model requires extensive grid search over β, M, T, and λ, which significantly increases computational cost. Since AP-OOD relies on an attention pooling mechanism, each configuration must be trained separately, making the search process expensive. This can limit scalability and practicality when applied to large datasets.

2. Although the paper claims that the unsupervised setting does not use AUX data for training, it still leverages AUX samples for hyperparameter selection. Moreover, the model’s performance appears highly sensitive to hyperparameter choices. This effectively makes the selection process semi-supervised and may introduce bias if the AUX distribution differs from the OOD distribution encountered at test time. As a result, the chosen model may not generalize reliably across unseen OOD scenarios.

**Questions:**

1. Attention weights indicate where the model is “looking,” but they do not necessarily reflect the tokens that causally contribute to the OOD decision. Could the authors empirically verify whether higher attention values actually correspond to greater influence on the final OOD score? Could the authors show if tokens with higher attention are indeed more important for the model’s OOD detection?

2. In Section A.2, the paper states that the Mahalanobis distance and the proposed decomposition are “equivalent”.  However, this statement seems mathematically ambiguous. It is unclear whether this equivalence truly holds for arbitrary linearly independent w_j. As currently written, the derivation seems to assume equivalence without sufficient justification. Could the authors explain this step in more detail?

3. The authors set the number of parameters in AP-OOD to match that of the Mahalanobis baseline for a fair comparison. However, it would be interesting to see how much further the performance could improve if the model were allowed to use a larger capacity.

4. AP-OOD employs multiple heads and multiple queries per head to capture diverse token-level informations. However, it remains unclear whether different heads and queries actually learn distinct OOD-related patterns or if they are largely redundant. Could the authors provide qualitative or quantitative evidence showing the contribution of each head to OOD detection? Such analysis would help clarify whether the attention structure learns meaningful diversity rather than merely increasing model capacity.

5. The authors mention that they ensure a fair comparison by matching the number of parameters between AP-OOD and the Mahalanobis baseline. However, this fairness applies only to model size, not necessarily to computational complexity or inference cost. It would be helpful to clarify how much additional computational overhead AP-OOD introduces in practice and whether this extra cost is justified by the observed performance improvements.

Please answer the questions in the rebuttal.

---

> ### Author Response · Authors · 2025-11-22
>
> **Response to Weaknesses**
>
> > The model requires extensive grid search over β, M, T, and λ, which significantly increases computational cost. Since AP-OOD relies on an attention pooling mechanism, each configuration must be trained separately, making the search process expensive. This can limit scalability and practicality when applied to large datasets.
>
> Hyperparameter tuning is a limitation of most approaches. However, as shown in Appendix D.8 in the updated manuscript, AP-OOD is relatively insensitive to the hyperparameter selection. In particular, on the summarization task in the Input OOD setting, AP-OOD improves the mean AUROC and mean FPR95 metrics over the Mahalanobis baseline in all tested settings for $\beta$, except for the setting where $\beta=0$. AP-OOD also surpasses the Mahalanobis baseline on these metrics across all tested settings for M and T on the summarization task.
>
> > Although the paper claims that the unsupervised setting does not use AUX data for training, it still leverages AUX samples for hyperparameter selection. Moreover, the model’s performance appears highly sensitive to hyperparameter choices. This effectively makes the selection process semi-supervised and may introduce bias if the AUX distribution differs from the OOD distribution encountered at test time. As a result, the chosen model may not generalize reliably across unseen OOD scenarios.
>
> There is no intrinsic reason for not leveraging AUX data for hyperparameter selection. As a matter of fact, it is common practice in OOD detection to assume access to some outlier data for hyperparameter selection --- even if the learning algorithm itself does not use auxiliary outlier data (e.g., Liang et al., 2018; Sun et al., 2022; Zhang et al., 2022; Lu et al., 2024). The reason for this is that in almost all settings, practitioners will have access to some outlier data to fit or to evaluate an OOD detector. That said, it is important that hyperparameters are selected on a separate set of outlier data, distinct from the OOD test data sets used for reporting results. Based on the amount and diversity of the available outlier data, practitioners can decide whether to use it only for model selection or for training a model in a semi- or fully supervised setting.

---

> ### Author Response · Authors · 2025-11-22
>
> **Response to Questions**
>
> > Attention weights indicate where the model is “looking,” but they do not necessarily reflect the tokens that causally contribute to the OOD decision. Could the authors empirically verify whether higher attention values actually correspond to greater influence on the final OOD score? Could the authors show if tokens with higher attention are indeed more important for the model’s OOD detection?
>
> To answer this question, we conducted the following experiment: We randomly select one sample from each of the four OOD data sets in the summarization benchmark. We then investigate the attention weights of a trained AP-OOD model over the generated output sequence. For each sample, we select the two heads with the largest deviations in the positive and in the negative directions before applying the square in the score function of AP-OOD. We then visualize the token-wise attention scores of the selected heads. We added the visualizations in Appendix D.4 in the updated manuscript. When we manually examined the generated output sequences, we found it hard to attribute the “OODness” of individual sequences to a single token or to a small set of tokens. Therefore, it is difficult to find a concrete interpretation of a head’s attention scores. However, we find that the different heads exhibit distinct attention patterns.
>
> > In Section A.2, the paper states that the Mahalanobis distance and the proposed decomposition are “equivalent”. However, this statement seems mathematically ambiguous. It is unclear whether this equivalence truly holds for arbitrary linearly independent w_j. As currently written, the derivation seems to assume equivalence without sufficient justification. Could the authors explain this step in more detail?
>
> By "equivalent", we meant the following correspondence:
>
> *1. Any linearly independent sequence $\(\boldsymbol{w}_1,\dots,\boldsymbol{w}_D\)$ induces a
> positive definite matrix*
>
> $$\boldsymbol{\Sigma}^{-1} := \sum_{j=1}^D \boldsymbol{w}_j \boldsymbol{w}_j^\top$$
>
> *and hence a Mahalanobis distance satisfying*
> \begin{equation}
> \sum_{j=1}^D (\boldsymbol{w}_j^\top \bar{\boldsymbol{z}}\ -\ \boldsymbol{w}_j^\top \boldsymbol{\mu})^2
> =\ (\bar{\boldsymbol{z}}\ -\ \boldsymbol{\mu})^\top \boldsymbol{\Sigma}^{-1} (\bar{\boldsymbol{z}}\ -\ \boldsymbol{\mu}).
> \tag{$\star$}
> \end{equation}
>
> *2. Conversely, any full-rank covariance matrix $(\boldsymbol{\Sigma})$ admits a
> set of linearly independent vectors such that
> $\boldsymbol{\Sigma}^{-1} = \sum_{j=1}^D \boldsymbol{w}_j \boldsymbol{w}_j^\top$, and therefore
> Eq. $(\star)$ holds.*
>
> Thus, our decomposition and the Mahalanobis form represent the same
> quadratic form; the eigen-decomposition shown in the appendix is only
> one possible choice of $\boldsymbol{w}_j$.
> For clarity, we have replaced the ambiguous original sentence with the
> explicit formulation above in the revised version.
>
>
> > The authors set the number of parameters in AP-OOD to match that of the Mahalanobis baseline for a fair comparison. However, it would be interesting to see how much further the performance could improve if the model were allowed to use a larger capacity.
>
> To answer this question, we conduct an additional scaling experiment on the summarization task (Input OOD): First, we train on all available ID examples (as opposed to the 100,000 used in the original manuscript), then we select the number of heads ($M$) from the set $\\{1, 16, 128, 1024\\}$  and the number of queries ($T$) from the set $\\{1, 4, 16\\}$. The parameter count of the largest configuration is approximately 17M, and 16 times larger than the configurations we tested in the original manuscript. We additionally select $\beta$ from the set $\\{1 / \sqrt{D}, 0.25, 0.5, 1, 2\\}$. For all configurations, we measure the mean AUROC on the OOD detection task. We find that higher values for $M$ and $T$ increase the mean AUROC, and the highest mean AUROC is attained by the largest configuration with $M=1024$, $T=16$, and $\beta=0.25$. Scaling beyond this point potentially improves the AUROC even more, but requires further optimization of our AP-OOD implementation for runtime and memory efficiency (e.g., using FlashAttention) or distributed training. We included the results for this experiment in the updated manuscript (Figure 4).

---

> ### Author Response · Authors · 2025-11-22
>
> > AP-OOD employs multiple heads and multiple queries per head to capture diverse token-level informations. However, it remains unclear whether different heads and queries actually learn distinct OOD-related patterns or if they are largely redundant. Could the authors provide qualitative or quantitative evidence showing the contribution of each head to OOD detection? Such analysis would help clarify whether the attention structure learns meaningful diversity rather than merely increasing model capacity.
>
> We find that heads are learnt largely independently from one another while queries are not, which we experimentally verify as follows: We train AP-OOD using the SGD optimizer on the summarization task using (i) 1 head and (ii) 2 heads, where the initialization of one of the heads in (ii) is identical to the initialization of the head of (i). We find that after training for 500 steps, the weight vectors associated with the heads with shared initialization between (i) and (ii) remain identical. In contrast, when repeating this experiment by varying the number of queries, the weight vectors associated with the queries differ after training. Intuitively, adding additional heads will help the model discover more local minima in the parameter space (similar to Lakshminarayanan et al., 2016), while adding queries increases the capacity of each given head. The following observation supports this intuition: When testing different hyperparameter combinations for AP-OOD, we found that a large number of queries combined with a small number of heads leads to overfitting when training the model on small ID data sets (e.g., 10,000 sequences): In this case, the average distance of the ID training sequences is substantially smaller than the average distance of the ID validation sequences. We added this discussion to Appendix C.3 of the revised manuscript.
>
> > The authors mention that they ensure a fair comparison by matching the number of parameters between AP-OOD and the Mahalanobis baseline. However, this fairness applies only to model size, not necessarily to computational complexity or inference cost. It would be helpful to clarify how much additional computational overhead AP-OOD introduces in practice and whether this extra cost is justified by the observed performance improvements.
>
> Indeed, while we match the Mahalanobis baseline in terms of parameter count for fairness, AP-OOD incurs computational overhead compared to the Mahalanobis method due to the attention pooling process. Compared to the Transformer encoder forward pass in the summarization task, the relative overhead of AP-OOD is less than 20% with our current implementation. So far, we have not extensively optimized our implementation for runtime performance, and we expect that AP-OOD can be made even more efficient by implementing Flash Attention (Dao et al., 2022). Compared with the computational cost of generating an output sequence in the summarization task, the relative overhead of AP-OOD is only 0.1%.
>
> Given this large runtime discrepancy, we find that the additional compute budget spent on AP-OOD can reduce the total computational burden compared to using the Mahalanobis baseline for OOD detection: Assume that the inference stream for the summarization task consists of 50% ID and 50% OOD examples and that the goal is to reject the OOD examples (i.e., to notify the user that no generation is possible). Additionally, assume that the threshold is set so that 5% of ID examples can be misclassified, and that the mean FPR95 of the OOD examples in the inference stream is identical to the mean FPR95 we reported in Table 1 of our paper. Given these assumptions, one can reject 60.18% of the OOD examples after an encoder forward pass with the Mahalanobis baseline, and one can reject 94.09% of the OOD examples after an encoder forward pass with AP-OOD. Additionally, we found that generating a summary for a sequence takes about 1.09s, applying the encoder followed by AP-OOD takes about 0.035s, and applying the encoder followed by Mahalanobis takes about 0.029s. In this setting, applying AP-OOD results in an expected 0.18s of wall-clock time savings per example (due to the generation time saved for the OOD examples correctly detected by AP-OOD). In addition, AP-OOD scales linearly with the context length, while the Transformer scales quadratically. Therefore, we expect greater compute savings as the context length increases. We have included detailed runtime analyses in Appendix D.9 of the updated manuscript.

---

> ### Author Response · Authors · 2025-11-22
>
> **References**
>
> Lakshminarayanan, Balaji, et al. "Simple and scalable predictive uncertainty estimation using deep ensembles." In NeurIPS 2017.
>
> Liang, Shiyu et al. "Enhancing The Reliability of Out-of-distribution Image Detection in Neural Networks." In ICLR 2018.
>
> Sun, Yiyou, et al. "Out-of-distribution detection with deep nearest neighbors." In ICML 2022.
>
> Zhang, Jinsong, et al. "Out-of-distribution detection based on in-distribution data patterns memorization with modern hopfield energy." In ICLR 2022.
>
> Dao, Tri, et al. "Flashattention: Fast and memory-efficient exact attention with io-awareness." In NeurIPS 2022.
>
> Lu, Haodong, et al. "Learning with mixture of prototypes for out-of-distribution detection." In ICLR 2024

---

### Official Review · Reviewer_18Vp · 2025-11-01

**Soundness:** 3
**Presentation:** 3
**Contribution:** 3
**Rating:** 8
**Confidence:** 3

**Summary:**

This paper points out the limitations of mean pooling for sequence representations and proposes Attention Pooling (AP-OOD) to compute OOD scores while preserving token-level information. The method uses attention (i) within each sequence and (ii) corpus-wide to estimate representative embeddings, then computes a distance/score between the input sequence’s token embeddings and the pooled corpus statistics.

During training, the backbone LM is frozen and only the AP-OOD module is trained; the approach applies seamlessly from unsupervised (ID-only) to semi-supervised settings using AUX data. The evaluation covers summarization (PEGASUS-LARGE trained on XSUM), translation (Transformer-base trained on WMT15 En–Fr), and audio (MIMII-DG), and reports both input-OOD (encoder embeddings) and output-OOD (decoder embeddings).

In experiment results, the paper shows improved AUROC and reduced FPR@95TPR, with especially large gains over embedding-based alternatives on summarization and audio.

**Strengths:**

- Replaces mean pooling with a learned attention-pooling formulation for OOD scoring; the paper clearly decomposes Mahalanobis into attention pooling and illustrates the failure mode of mean pooling with an intuitive toy example.

- Broad empirical coverage across two NLG tasks and one audio task; AP-OOD improves AUROC/FPR95 in unsupervised, semi-supervised, and supervised regimes, with consistent AUX scaling curves. The audio experiment demonstrates modality generality (MIMII-DG).

**Weaknesses:**

- (W1) Scope limited to task specific backbone models:
The models used (PEGASUS-LARGE or Transformer-base) do not consider modern LLMs (e.g., Llama, Qwen, Phi, etc.). Even if one assumes task-specific language models, today’s practice allows fine-tuning LLMs or few-shot (in-context) learning. Defining ID vs. OOD simply as “training data vs. domain-shifted datasets” can be somewhat unrealistic for contemporary LLM usage.

- (W2) Does not directly address hallucination:
Hallucination is a central issue in language generation. While OOD detection is related to hallucination mitigation, the paper offers no empirical evidence or discussion of how AP-OOD might help reduce hallucinations.

**Questions:**

**Please provide responses to the reviews mentioned in the Weaknesses section.**

**Additional questions**

(Q1) Application to modern LLMs & hallucination mitigation:
- Why wasn’t AP-OOD validated on advanced open-source LLMs such as Llama, Qwen, or Phi?

(Q2) Input-/Output-OOD for unsafe or malicious prompts:
- (Q2-1) How should AUX be constructed to enable reliable detection of unsafe prompts (e.g., jailbreaks or prompts that solicit illegal activity)?
- (Q2-2) Based on the method and the reported results, do you expect AP-OOD to be effective for blocking malicious inputs (and, by extension, for detecting unsafe outputs)? If so, under what conditions?

---

> ### Author Response · Authors · 2025-11-22
>
> **Response to Weaknesses**
>
> > (W1) Scope limited to task specific backbone models: The models used (PEGASUS-LARGE or Transformer-base) do not consider modern LLMs (e.g., Llama, Qwen, Phi, etc.). Even if one assumes task-specific language models, today’s practice allows fine-tuning LLMs or few-shot (in-context) learning. Defining ID vs. OOD simply as “training data vs. domain-shifted datasets” can be somewhat unrealistic for contemporary LLM usage.
>
> We agree with this assessment. Our contribution follows the experimental setup for OOD detection for language modeling proposed by Ren et al. (2023) to make a comparison possible. We see three main challenges in applying this setup to contemporary LLMs:
> 1. The setting assumes access to the model’s training data for fitting the OOD detector. However, the training data sets for many LLMs (including the three LLMs mentioned by the reviewer) are not publicly available.
> 2. Many LLMs are trained on large amounts of text. This makes it difficult to define appropriate data sets that are fully OOD w.r.t. the training data for evaluation. Finding a large and diverse auxiliary outlier data set for training an OOD detector in the supervised setting is even more challenging.
> 3. The training process of LLMs usually involves multiple stages (pre-training, instruction tuning, RLHF, …), making it difficult to pinpoint an exact training distribution $p_\text{ID}$ for an LLM.
>
> For these reasons, we consider OOD detection for modern LLM pipelines as outside the scope of our work.
>
> However, we have included a discussion regarding LLMs in the Future Work section of the updated manuscript. And, to verify the general applicability of AP-OOD to the decoder-only language modeling paradigm of LLMs, in addition to the existing summarization and translation tasks, we now conduct experiments on Pythia-160M (a decoder-only LM trained on the Pile, a large open-source data set of English text). We have included the experiments and results in Appendix D.6 in the updated manuscript. To summarize, we compare OOD detection results across five OOD test data sets and find that AP-OOD outperforms the second-best method (Perplexity) in terms of both mean AUROC (from 77.59% to 86.67%) and mean FPR95 (from 59.25% to 55.81%).
>
> > (W2) Does not directly address hallucination: Hallucination is a central issue in language generation. While OOD detection is related to hallucination mitigation, the paper offers no empirical evidence or discussion of how AP-OOD might help reduce hallucinations.
>
> We agree with the reviewer that we should have made the connection more explicit to benefit the paper (in the original manuscript we only briefly alluded to it in the related work section). Therefore, we have updated the manuscript to include the following paragraph in the introduction:
>
> Supplying OOD prompts to a language model can result in model hallucination - i.e., the model generating output that is nonsensical or unfaithful to the prompt (Farquhar et al., 2024). For example, Ren et al. (2023) observe that a common failure case in abstractive summarization is for the model to output “All images are copyrighted” when prompted to summarize news articles from a publisher (CNN) that differs from what it was trained on (BBC). In many works, hallucination is attributed to model uncertainty (e.g., Farquhar et al., 2024; Aichberger et al., 2025), which can be decomposed into aleatoric uncertainty (resulting from noise in the data) and epistemic uncertainty (resulting from a lack of training data). OOD prompts exhibit high epistemic uncertainty (Ren et al., 2023).

---

> ### Author Response · Authors · 2025-11-22
>
> **Response to Questions**
>
> > (Q1) Why wasn’t AP-OOD validated on advanced open-source LLMs such as Llama, Qwen, or Phi?
>
> In the original version of the manuscript, we focus on the benchmark setting proposed by Ren et al. (2023), which focuses on OOD detection for translation and summarization tasks. Applying OOD detection to LLMs such as Llama, Qwen, or Phi involves several challenges (see response to W1), which we regard as outside the scope of this work. However, to validate the general applicability of our method to the decoder-only modeling paradigm of LLMs, we now also evaluate AP-OOD on language modeling using Pythia-160M on the Pile as the ID data set. The results are consistent with our previous analysis: AP-OOD improves OOD detection on the language modeling task.
>
> > (Q2-1) How should AUX be constructed to enable reliable detection of unsafe prompts (e.g., jailbreaks or prompts that solicit illegal activity)?
>
> Detecting unsafe prompts is a crucial problem for the reliability of language models. We find the reviewer's suggestion interesting and therefore consider detecting unsafe prompts an exciting avenue for future work. AP-OOD can potentially be used for this use case when trained in the supervised setting with a large and diverse collection of unsafe prompts as the auxiliary data set.
> > (Q2-2) Based on the method and the reported results, do you expect AP-OOD to be effective for blocking malicious inputs (and, by extension, for detecting unsafe outputs)? If so, under what conditions?
>
> It is hard to say with certainty how effective AP-OOD is for detecting malicious prompts. However, we believe that it might be effective when:
> - AP-OOD is trained in the supervised setting with an appropriate auxiliary data set (see answer to Q2-1)
> - The ID data contains no or only a very small number of unsafe inputs
> - It is a desirable (or at least a non-detrimental) property that inputs are also flagged as unsafe when they are “only” OOD.
>
> **References**
>
> Ren, Jie, et al. "Out-of-distribution detection and selective generation for conditional language models." In ICLR 2023
>
> Farquhar, Sebastian, et al. "Detecting hallucinations in large language models using semantic entropy." In Nature 630.8017 (2024): 625-630.
>
> Aichberger, Lukas, et al. "Improving uncertainty estimation through semantically diverse language generation." In ICLR 2025

---

### Author Response · Authors · 2025-11-22
**General response**

We thank all reviewers for providing insightful feedback and high-quality reviews. It allowed us to significantly enhance our paper. In summary, the reviewers appreciate
- the clear motivation and intuitive illustration of the failure mode of mean pooling (reviewers `18Vp`, `WMzy`, `QF7M`), as well as
- the meaningful empirical improvements in OOD detection in the natural language and audio tasks (reviewers `18Vp`, `M9r8`, `QF7M`).

The reviewers offered divergent assessments of the manuscript's clarity. While reviewer `M9r8` found that "the paper suffers from clarity and presentation issues", reviewer `QF7M` commended that "the paper presents a very clear motivation". Reviewers `18Vp` and `QF7M` were interested in how AP-OOD can be applied to large language models (LLMs). We have now added a discussion on LLMs in the future work section of the updated manuscript and included an experiment on the Pythia language model to showcase the applicability of AP-OOD on the decoder-only language modeling paradigm of LLMs in Appendix D.6. Thanks to the feedback we received from reviewers, we were also able to substantially strengthen the presentation of our manuscript and the experimental evaluation of AP-OOD. In summary, we have made the following additions and improvements:

- We added a paragraph in the introduction that discusses the connection between OOD prompts and the problem of hallucination.
- We included PyTorch/Einops-like pseudocode implementing the loss computation of AP-OOD in Appendix C.2.
- We conducted a scalability study, which increases the capacity of AP-OOD beyond the setting we used in the original manuscript (matching the number of parameters of the Mahalanobis baseline) in the experimental section (Figure 4).
- We conducted a runtime analysis of AP-OOD, demonstrating that applying AP-OOD results in a small overhead compared to a standard Transformer encoder forward pass (Appendix D.9).
- We included experiments on OOD detection on Pythia-160M, trained on the Pile as the ID dataset. Overall, we find that AP-OOD yields the best results (Appendix D.6).
- We experimentally investigated how the model assigns attention weights to the individual tokens and added corresponding visualizations in Appendix D.4.
- We updated the derivation of the Mahalanobis decomposition in Appendix B.2 and provided a more rigorous definition of the relationship between the decomposed and matrix representations of the Mahalanobis distance.
- We added additional motivation for the semi-supervised setting with a practical, relevant example.
- We increased the prominence of the mini-batch attention pooling algorithm (Appendix C.1) as an efficient method for computing attention pooling over all tokens in the corpus ($\mathbf{\tilde{Z}}$).

We included the updated manuscript in our submission and color-coded the updated sections to facilitate the review.

---

### Author Response · Authors · 2025-11-25

Dear Reviewers,

Thank you again for your time and insightful feedback.

We have uploaded a revised manuscript and believe all points raised in your reviews have been fully addressed. We would be pleased to engage in further discussion should you require additional clarification.

We kindly ask you to consider the updated paper and hope that, if you feel your concerns have been adequately addressed, these revisions will be reflected in your final assessment.

Sincerely,

The Authors

---

### Meta-Review · Area_Chair_ViV5 · 2025-12-29

**Summary:**

The paper proposes AP-OOD, a method for out-of-distribution (OOD) detection in natural language processing. The core contribution is replacing standard mean pooling of token embeddings—which often obscures token-level anomalies—with an attention-based pooling mechanism grounded in a decomposition of the Mahalanobis distance. The method supports both unsupervised and semi-supervised settings.

Reviewers generally praised the clear motivation (the failure mode of mean pooling), the intuitive toy experiments, and the strong empirical performance on summarization and translation tasks. Initial concerns primarily focused on the scope being limited to encoder-decoder architectures (missing modern decoder-only LLMs), presentation clarity/mathematical rigor, and questions regarding computational overhead.

The authors provided a comprehensive rebuttal, notably adding experiments with a decoder-only LLM (Pythia-160M), clarifying the mathematical derivations, and providing runtime analyses. Given the solid theoretical grounding and the effective response to the critique regarding modern architecture applicability, the AC recommends accepting this paper.

**Reviewer Concerns:**

Concerns Addressed by the Rebuttal:
- Applicability to LLMs (Reviewers 18Vp, QF7M): The most significant concern was that the method was only tested on older BERT/PEGASUS-style architectures. The authors added Appendix D.6, demonstrating AP-OOD's effectiveness on Pythia-160M (a decoder-only model), showing it outperforms perplexity-based baselines.
- Presentation and Mathematical Clarity (Reviewers M9r8, WMzy): Reviewers noted undefined symbols and ambiguous mathematical claims (specifically the equivalence between Mahalanobis and the proposed decomposition). The authors revised the text to explicitly define the correspondence between the decomposition and the quadratic form, and added PyTorch-style pseudocode to clarify the implementation.
- Computational Overhead and Scalability (Reviewer WMzy): Concerns regarding the cost of the attention mechanism and hyperparameter search were addressed via a new runtime analysis (showing <20% overhead relative to the encoder and negligible overhead relative to generation) and a scalability study (Figure 4) demonstrating performance gains with increased heads/queries.
- Memory Constraints (Reviewer M9r8): The concern that concatenating embeddings is infeasible for large datasets was addressed by highlighting and clarifying the mini-batch attention pooling algorithm (Algorithm 2/Appendix C.1), which avoids loading the full corpus into memory.
- Missing Baselines (Reviewer M9r8): The authors clarified that the suggested baselines (DICE, VIM, GradNorm) are designed for classifier-based architectures, whereas this work targets generative language models, justifying their exclusion.

Outstanding Concerns:
- Novelty (Reviewer QF7M): While the authors argued that integrating attention with Mahalanobis via directional decomposition is novel, this reviewer may still view the combination of existing components (Attention + Mahalanobis) as incremental.
- Definition of "Unsupervised" (Reviewer WMzy): The concern that using auxiliary (AUX) data for hyperparameter tuning technically makes the method semi-supervised remains valid, though the authors correctly noted this is standard practice in the OOD literature.

**Reviewer Scores:**

Reviewer Scores
- Reviewer 18Vp (Score: 8 -> 8/9): This reviewer was already very positive, giving an "Accept" (8). Their main weakness was the lack of LLM validation. Since the authors added the Pythia-160M experiments, this reviewer would likely maintain their high score or potentially raise it to a 9, seeing their primary critique fully resolved.
- Reviewer WMzy (Score: 6 -> 7): This reviewer was on the fence. The authors provided extensive data to answer their questions regarding attention visualization, runtime overhead, and mathematical clarity. With the scalability and runtime concerns addressed, this reviewer would likely move their score up to a solid Accept (7).
- Reviewer M9r8 (Score: 2 -> 4/5): This reviewer recommended rejection based largely on "clarity and presentation issues" and "hasty writing." The authors revised the manuscript significantly, added pseudocode, and clarified the memory efficiency (mini-batching). While the reviewer might still prefer mixture models over the Gaussian assumption, the substantial improvements to presentation and clarity should move this score from a strong reject to a borderline or weak reject (4 or 5).
- Reviewer QF7M (Score: 4 -> 5/6): This reviewer felt the paper was marginally below the threshold due to a lack of "groundbreaking" novelty and lack of LLM experiments. The addition of the Pythia experiments directly addresses the applicability concern. While the novelty concern is subjective, the demonstrated effectiveness on the new architecture makes the paper more robust, likely pushing this reviewer to a weak accept (5 or 6).

---

### Decision · Program_Chairs · 2026-01-26

Accept (Poster)